# RHEB neddylation by the UBE2F-SAG axis enhances mTORC1 activity and aggravates liver tumorigenesis

Fengwu Zhang[1,2,3], Xiufang Xiong[1,2,3], Zhijian Li [1,2,3], Haibo Wang [1,2,3], Weilin Wang[1,2], Yongchao Zhao [3,4,5✉] & Yi Sun [1,2,3,5,6,7,8✉]

## Abstract

**Small GTPase RHEB is a well-known mTORC1 activator, whereas neddylation modifies cullins and non-cullin substrates to regulate their activity, subcellular localization and stability. Whether and how RHEB is subjected to neddylation modification remains unknown. Here, we report that RHEB is a substrate of NEDD8-conjugating E2 enzyme UBE2F. In cell culture, UBE2F depletion inactivates mTORC1, inhibiting cell cycle progression, cell growth and inducing autophagy. Mechanistically, UBE2F cooperates with E3 ligase SAG in neddylation of RHEB at K169 to enhance its lysosome localization and GTP-binding affinity. Furthermore, liver-specific *Ube2f* knockout attenuates steatosis and tumorigenesis induced by *Pten* loss in an mTORC1-dependent manner, suggesting a causal role of UBE2F in liver tumorigenesis. Finally, UBE2F expression levels and mTORC1 activity correlate with patient survival in hepatocellular carcinoma. Collectively, our study identifies RHEB as neddylation substrate of the UBE2F-SAG axis, and highlights the UBE2F-SAG axis as a potential target for the treatment of non-alcoholic fatty liver disease and hepatocellular carcinoma.**

**Keywords** RHEB; UBE2F; mTORC1; Neddylation; Liver Steatosis and Tumorigenesis
**Subject Categories** Cancer; Signal Transduction

## Introduction

Neddylation is a crucial post-translational modification, through which NEDD8, a ubiquitin-like molecule is attached to the lysine residue of a substrate protein. Like ubiquitylation, neddylation is catalyzed by a cascade of enzymes, including NEDD8-activating enzyme E1 (NAE), NEDD8-conjugating enzyme E2s (NCE), and substrate-specific NEDD8-E3 ligases (Zhao et al, 2014). In mammalian cells, there are only two members of NCE family, namely UBE2M (also known as UBC12) and UBE2F. UBE2M pairs with RBX1 E3 to promote neddylation of cullins 1-4, leading to activation of CRLs1-4 (Cullin-RING ligases), while UBE2F pairs with SAG/RBX2 to neddylate cullin-5, resulting in CRL5 activation (Zhao et al, 2020). As the largest family of E3 ubiquitin ligase, CRLs promote ubiquitylation and subsequent proteasome degradation of many key cellular proteins to maintain the homeostasis of a cell (Zhang et al, 2024; Zhou et al, 2018a).

Abnormal activation of neddylation has been frequently found in many types of human diseases and human cancers (Zhang et al, 2024; Zhou et al, 2018a). More specifically, over-activation of neddylation cascade was found in a number of liver diseases. For example, cullin-mediated degradation of AACA2 (acetyl-CoA acyltransferase 2) promoted liver steatosis, whereas CAND1 (Cullin-associated and neddylation-dissociated protein 1), which blocks cullin neddylation (Zheng et al, 2002), inhibited ACAA2-induced liver lipid accumulation (Huang et al, 2023). Furthermore, inhibition of neddylation has been shown to reduce fibrosis by promoting c-Jun-mediated apoptosis in hepatic stellate cells (HSCs) (Zubiete-Franco et al, 2017). On the other hand, liver specific knockout of NAE1, the regulatory subunit of the only NEDD8 E1 enzyme, triggered hepatocyte death, inflammation, and fibrosis, leading to severe liver damage, which can be partially alleviated by inhibiting the accumulation of NIK (NF-κB-inducing kinase), a neddylation substrate (Xu et al, 2022a). However, whether and how UBE2F, one of neddylation E2s, regulates the growth of liver cells in vitro and affects liver tumorigenesis in vivo are previously unknown.

The mechanistic target of rapamycin complex 1 (mTORC1) consists of mTOR, RAPTOR, DEPTOR, GβL, and PRAS40, of which the RAPTOR is essential for mTORC1 localization and substrate recruitment (Aylett et al, 2016; Condon and Sabatini, 2019). As a master regulator of cell growth, the mTORC1 integrates environmental signals to control the balance between anabolic and catabolic metabolism (Laplante and Sabatini, 2012). Upon activation, mTORC1 phosphorylates a number of proteins, including 4E-BP1, S6K1, TFEB1, and ULK1 to facilitate the synthesis of protein, lipid, and nucleotide and inhibition of autophagy to maintain the homeostasis of a cell (Liu and Sabatini, 2020). Importantly, the

[1]Cancer Institute, The Second Affiliated Hospital, Zhejiang University School of Medicine, 310009 Hangzhou, China. [2]Key Laboratory of Cancer Prevention and Intervention, China National Ministry of Education, 310009 Hangzhou, China. [3]Institute of Translational Medicine, Zhejiang University School of Medicine, 310029 Hangzhou, China. [4]Department of Hepatobiliary and Pancreatic Surgery, The First Affiliated Hospital, Zhejiang University School of Medicine, Hangzhou, China. [5]Cancer Center, Zhejiang University, 310058 Hangzhou, China. [6]Leading Innovative and Entrepreneur Team Introduction Program of Zhejiang, Hangzhou, China. [7]Research Center for Life Science and Human Health, Binjiang Institute of Zhejiang University, 310053 Hangzhou, China. [8]Institute of Fundamental and Transdisciplinary Research Zhejiang University, Hangzhou, China. ✉E-mail: yongchao@zju.edu.cn; yisun@zju.edu.cn

mTORC1 pathway is essential to fulfill the heightened energy requirements for proliferating cancer cells. Notably, this pathway is upregulated in approximately 50% of hepatocellular carcinomas (HCCs), and associated with dysregulation of the phosphatase and tensin homolog (PTEN). This dysregulation underscores the significance of the mTORC1 pathway in the pathogenesis of HCC and highlights its potential as a therapeutic target (Villanueva et al, 2008; Bhat et al, 2013).

Activation of mTORC1 occurs at the cytoplasmic face of the lysosomal membrane and is mediated by two groups of small GTPases, the RAGs and RHEB (Ben-Sahra and Manning, 2017). RAGs localize in lysosome and bind to RAPTOR upon amino acid stimulation, leading to the recruitment of mTORC1 from the cytosol to the lysosome, which allows lysosomal RHEB to stimulate mTORC1 kinase activity (Sancak et al, 2008). Indeed, RHEB GTPase directly activates mTORC1, and RHEB overexpression causes constitutive activation of mTOR even in the absence of nutrients (Sato et al, 2009). Cryo-EM structures revealed that RHEB activates mTORC1 by allosterically realigning active-site residues, bringing them into the correct conformation for catalysis (Yang et al, 2017). Growth factors, hypoxia, and energy stress regulates mTORC1 activity via modulating the TSC-RHEB axis (Brugarolas et al, 2004; Inoki et al, 2003b). The TSC1-TSC2-TBC1D7 triple complex acts as a GTPase-activating protein (GAP) for RHEB (Dibble et al, 2012; Yang et al, 2021). Upon insulin stimulation, insulin/insulin-like growth factor 1 (IGF-1) activates AKT, which phosphorylates TSC2 at the multiple sites to dissociate TSC from the lysosomal surface, leading to mTORC1 activation (Garami et al, 2003; Manning et al, 2002). Thus, maximal mTORC1 activation dependent on both intracellular and extra-cellular signals.

Several studies showed that RHEB is subjected to post-translational modification by ubiquitylation. Lysosome-anchored E3 ligase RNF152 promotes the mono-ubiquitylation of RHEB on K8 site to enhance the binding of RHEB with TSC complex for RHEB inactivation, a process reversed by deubiquitylase USP4 (Deng et al, 2019). Additionally, it was reported that amino acids promote the polyubiquitylation of lysosomal RHEB to enhance its binding with mTORC1, a process blocked by lysosomal deubiquitylase ATXN3 (Ataxin 3) (Yao et al, 2020). However, whether and how RHEB is subjected to neddylation modification and its biochemical/biological consequences are completely unknown.

In this study, we reported that RHEB is subjected to neddylation modification on K169, catalyzed by the UBE2F-SAG E2-E3 pair, which enhances RHEB activity and lysosomal localization to activate mTORC1. Biochemically, knockdown of UBE2F or SAG inactivates mTORC1 via reducing RHEB neddylation. Biologically, knockdown of UBE2F or SAG suppresses growth, cell cycle progression, reduced cell size, and induces autophagy in in vitro liver cancer cell culture models, whereas *Ube2f* liver specific deletion inhibits steatosis and tumorigenesis in an in vivo liver cancer model triggered by *Pten* loss. Finally, a positive correlation between UBE2F levels and mTORC1 activity (pS6) was observed in human liver cancer tissues. Thus, our study revealed a novel mechanism by which neddylation activates the RHEB-mTORC1 signaling pathway, and provided proof-of-concept evidence that the UBE2F-SAG axis could be an attractive anti-liver cancer target.

# Results

## UBE2F is overexpressed in liver cancer tissues, which correlates with poor patient survival

Our previous study showed that UBE2F, one of two known neddylation E2 conjugating enzymes, is overexpressed in non-small cell lung carcinomas and protects lung cancer cells from apoptosis by activating CRL5 (cullin-RING ligase-5) to promote ubiquitylation and degradation of proapoptotic protein NOXA (Zhou et al, 2017). Whether and how UBE2F is altered in liver cancer to regulate cell growth and survival and to affect liver tumorigenesis are previously unknown.

We first analyzed the TCGA database and found that the *UBE2F* mRNA level was overexpressed in liver hepatocellular carcinoma (LIHC) tumor tissues, as compared to normal liver tissues (Appendix Fig. S1A). More importantly, *UBE2F* overexpression correlated significantly with poor patient survival (Appendix Fig. S1B). We then used liver tumor tissue microarray, and found that the UBE2F staining were also significantly higher in tumor tissues, as compared with adjacent normal tissues (Fig. 1A,B; Appendix Fig. S1C), and again higher UBE2F protein levels correlated with a worse survival of liver cancer patients (Fig. 1C). Collectively, these correlation studies highly suggest that UBE2F is involved in and may play a promoting role during liver tumorigenesis.

## UBE2F knockdown suppresses growth and survival and induces autophagy

We next investigated the role of UBE2F in growth regulation of liver cancer cells using the in vitro cell culture models with a loss-of-function approach. We found that UBE2F knockdown significantly inhibited the growth/proliferation, and clonal survival of PLC/PRF/5 and SK-HEP-1 liver cancer cells (Figs. 1D–F and EV1A,B). To define the nature of reduced growth/proliferation, we first used BrdU-incorporation assay, and found that UBE2F knockdown significantly slowed the S phase entry (Fig. 1F; Appendix Fig. S1D). We further used double thymidine block assay to synchronize cells at the G1/S boundary. The release experiment showed that UBE2F knockdown caused cell arrest at both G1 and G2/M phases of cell cycle with increased levels of p21 (for G1 phase) and cyclin B1 (for G2/M phase) (Figs. EV1C–F; Appendix Fig. S1E–H). We then measured possible apoptosis induction, using cleaved caspase-3 and PARP as the readouts, given our previous study showing that UBE2F knockdown induced robust apoptosis in lung cancer cells via causing NOXA accumulation (Zhou et al, 2017). Among three liver cancer cell lines tested, UBE2F knockdown induced modest apoptosis in PLC/PRF/5 cells, but not at all in SK-HEP-1 and Hep3B cells (Fig. EV1G). Thus, the major cause of growth inhibition upon UBE2F knockdown was not due to apoptosis in liver cancer cells. Finally, we found that UBE2F knockdown significantly reduced cell size (Fig. 1G,H) and induced autophagy with increased LC3-II conversion and reduced p62 levels (Figs. 1I–K and EV1H,I). Moreover, treatment of liver cancer cells with 3-MA, an inhibitor of autophagy induction, caused significant alleviation of LC3-II accumulation triggered by UBE2F knockdown (Fig. 1L), indicating a causal involvement of autophagy induction. Taken together,

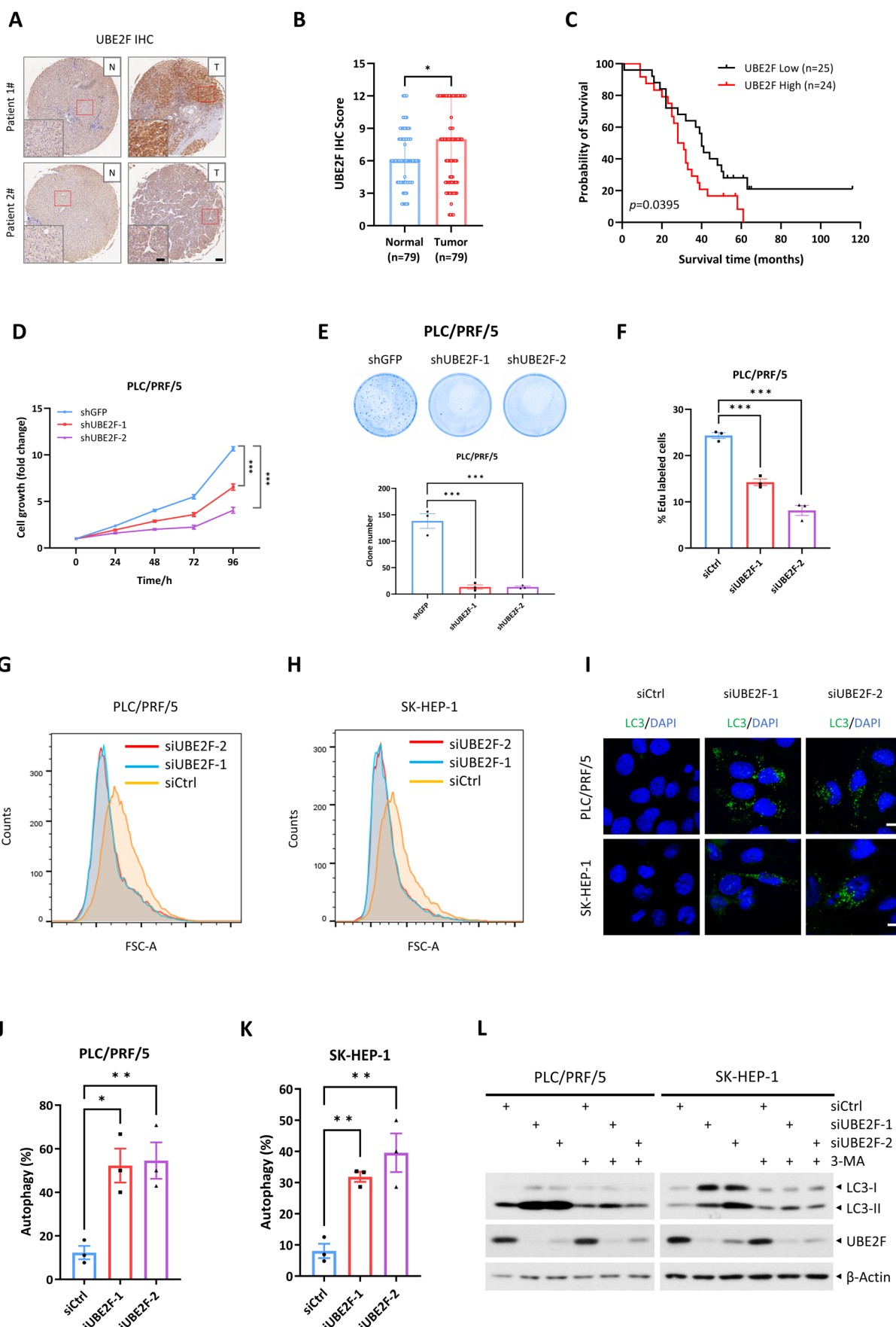

**Figure 1.  High UBE2F expression correlates with poor survival of liver cancer patients and UBE2F knockdown suppresses growth and survival by inducing autophagy.**

(A, B) The liver tumor tissue microarrays containing 79 tumor tissues (T) and adjacent normal tissues (N) were stained for UBE2F levels, and the representative images of UBE2F staining are shown (A). The immunohistochemical (IHC) staining scores of UBE2F in liver tumor tissue microarrays (A) were analyzed (B). The insets show enlarged images of the red boxes. Scale bars, 120 μm and 60 μm (inset), respectively. Data were presented as median with interquartile range and analyzed by Student's $t$ test ($n = 79$, $P = 0.0478$). *$P < 0.05$. (C) The Kaplan–Meier survival analysis (using the Mantel–Cox log-rank test) of liver cancer patients with high and low UBE2F expression in (A) are shown. The IHC score of UBE2F greater than or equal to 9 is designated as high expression, while the IHC score less than or equal to 4.5 is designated as low expression. (D, E) PLC/PRF/5 cells stably expressing shRNA targeting UBE2F were subjected to cell growth assays (D) and clonogenic assays (E). Representative images of the clonogenic assays are shown (top, E), and the colony numbers are plotted (bottom, E). Data were shown as mean ± SEM from three independent experiments, and analyzed by two-way ANOVA (D) and one-way ANOVA (E), respectively. The $P$ values were 8.3E-14 for siCtrl vs. siUBE2F-1 and 5E-14 for siCtrl vs. siUBE2F-2 (D). For (E), the $P$ values were 7.95E-5 for both siCtrl vs. siUBE2F-1 and siCtrl vs. siUBE2F-2. ***$P < 0.001$. (F) PLC/PRF/5 cells were transfected with the indicated siRNAs for 48 h and then labeled with 20 μM EdU for 2 h. The cells were subsequently fixed and incubated with Azide 488 before being subjected to flow cytometry. The percentages of proliferating cells were shown as mean ± SEM from three independent experiments and analyzed by one-way ANOVA (siCtrl vs. siUBE2F-1, $P = 2.57E-4$; siCtrl vs. siUBE2F-2, $P = 1.70E-5$). ***$P < 0.001$. (G, H) PLC/PRF/5 (G) and SK-HEP-1 (H) cells were transfected with the indicated siRNAs for 48 h. Cells were then trypsinized and resuspended in PBS, followed by flow cytometry to assess cell size. (I–K) Cells were transfected with indicated siRNAs for 72 h, and then stained with anti-LC3 antibody, followed by photography under a confocal fluorescent microscope. Representative images (I) and the statistical analysis of the percentages of cells exhibiting LC3-positive puncta (J, K) are shown. Data were shown as mean ± SEM from three independent experiments and analyzed by one-way ANOVA. The $P$ values were 0.0109 (siCtrl vs. siUBE2F-1) and 0.0084 (siCtrl vs. siUBE2F-2) for (J); 0.0092 (siCtrl vs. siUBE2F-1) and 0.0023 (siCtrl vs. siUBE2F-2) for (K). *$P < 0.05$; **$P < 0.01$. Scale bars, 10 μm. (L) Cells were transfected with indicated siRNAs for 24 h, and then treated with 10 mM 3-MA or vehicle for an additional 48 h, followed by immunoblotting (IB) analysis using the indicated antibodies (Abs). Source data are available online for this figure.

UBE2F knockdown inhibits both cell growth and cell proliferation as well as induces autophagy.

## UBE2F knockdown inactivates mTORC1 activity

Given the mTORC1 signal pathway is a well-known key regulator of cell size and autophagy (Kim and Guan, 2019), we next determined the effect of UBE2F on activities of both mTORC1 and mTORC2 as well as MAPK, yet another predominant signaling pathway that regulates growth/proliferation and survival. Strikingly, UBE2F knockdown significantly inactivated mTORC1 activity, as reflected by remarkable reduction in the levels of p4E-BP1 and pS6K1, as well as the ratio of p4E-BP1/4E-BP1 and pS6K1/S6K1 in two lines of liver cancer cells (Figs. 2A and EV2A–D). On the other hand, UBE2F knockdown had minor, if any, effect on the levels of pAKT/mTORC2 and MAPK/pERK in a cell line-dependent manner (Fig. 2A). Thus, UBE2F mainly regulates the mTORC1 activity with minimal, if any, effect on the mTORC2/AKT and RAS/ERK pathways. We, therefore, focused our attention to understand how UBE2F regulates the mTORC1 signal for the rest of our study.

We next determined whether this effect is specific for UBE2F, not for its family member UBE2M. Indeed, unlike UBE2F knockdown, UBE2M knockdown resulted in moderate increase or decrease in the levels of pS6K1 and p4E-BP1 in a cell line dependent manner among three liver cancer cell lines and HeLa cells (Figs. 2B and EV2E–G). Thus, UBE2M is unlikely a bona fide mTORC1 regulator. More interestingly, UBE2F knockdown-induced mTORC1 inactivation was in a manner independent of growth factors, since the same phenomenon was observed in all three lines of liver cancer cells after 24 h of serum starvation (Fig. EV2H), indicating that UBE2F regulates the intrinsic activity of mTORC1. Again, UBE2M knockdown had no effect on the mTORC1 signal, regardless of serum levels (Figs. 2B and EV2E–H).

To examine whether mTORC1 inactivation by UBE2F knockdown is a general phenomenon, we used two pairs of mouse embryonic fibroblasts (MEFs), and found that $Ube2f$ knockout indeed inactivated the mTorc1 activity as well (Fig. 2C). We further used MEFs to investigate the impact of Ube2f on mTorc1 activity

under conditions of serum starvation followed by serum refeeding. Strikingly, $Ube2f$ knockout not only dampened the basal levels of mTorc1 signaling, but also impeded mTorc1 activation induced by growth factors (Fig. 2D). Finally, using an in vitro kinase assay, we found that UBE2F knockdown inhibited in vitro phosphorylation of S6K1 by mTORC1 (Fig. 2E), strongly suggesting that UBE2F is a bona fide mTORC1 activator.

We then determined whether UBE2F is one of previously unrealized components of mTORC1 complex whose absence (via knockdown or knockout) would inactivate mTORC1 activity. We used a gel filtration chromatography assay, and found that mTORC1-associated components, such as RAPTOR and GβL were enriched within fractions 19 to 23, whereas UBE2F were detectable in much latter fractions 32-35, indicating UBE2F does not assembly with mTORC1 complex (Fig. EV2I). Furthermore, UBE2F was not detected in an active mTORC1 complex by an immunoprecipitation assay (Fig. 2E). Collectively, these results highly suggested that UBE2F likely affects an mTORC1 upstream regulator(s). In fact, we found that UBE2F knockdown did not significantly affect the levels of few upstream regulators of mTORC1, including HIF1α, DEPTOR, TSC1, or TSC2 (Cui et al, 2016), although REDD1 levels were reduced in a cell line-dependent manner (Fig. EV2J). We reasoned that these minor reductions could be attributable to the suppression of global protein translation, resulting from mTORC1 inactivation. Collectively, it appears that UBE2F regulates mTORC1 activity independent of direct interaction with the mTORC1, nor through degradation of mTORC1 upstream regulators by CRL5 ubiquitin ligase, which is activated by UBE2F (Zhang et al, 2024).

## The UBE2F-SAG axis regulates mTORC1 activity in a RHEB-dependent manner

We then focused on neddylation mechanism on mTORC1 regulation. First, we confirmed that like UBE2F, siRNA-based knockdown of SAG, a UBE2F-paired neddylation E3, but not RBX1, a UBE2M-paired E3, nor CUL-5, a neddylation substrate of the UBE2F-SAG pair, inactivated mTORC1 activity (Fig. 3A), indicating that the regulation occurs at the level of neddylation by

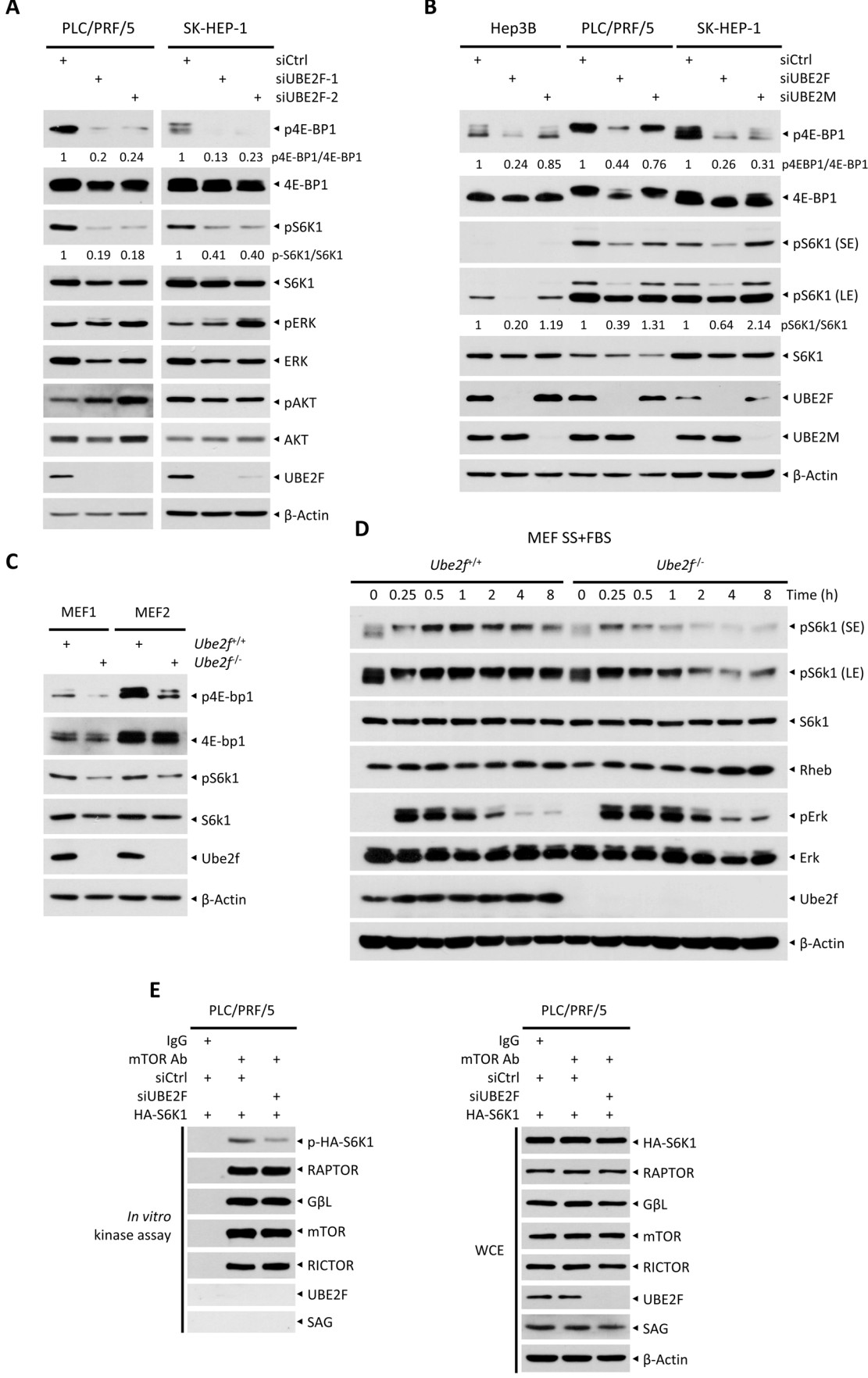

**Figure 2.  UBE2F knockdown inactivates mTORC1 activity.**

(A) Cells were transfected with indicated siRNAs for 72 h, followed by IB analysis. (B) Cells were transfected with indicated siRNAs for 72 h and then harvested for IB with indicated Abs. (C, D) *Ube2f*fl/fl MEF cells were infected with Ad-GFP or Ad-Cre adenovirus for 72 h. After infection, cells were harvested for IB analysis (C), or subjected to serum starvation for 24 h, followed by serum re-supply for indicated time periods before being harvested for IB analysis with indicated Abs (D). (E) PLC/PRF/5 cells were transfected with siRNA targeting UBE2F or control siRNA (siCtrl) for 48 h, followed by immunoprecipitation (IP) with mTOR Ab, or normal IgG as a negative control. The immunoprecipitated mTOR complexes were then incubated with unphosphorylated purified HA-S6K1 protein, followed by in vitro kinase assay (left). The whole cell extract (WCE) of PLC/PRF/5 cells for IP were subjected to IB analysis with indicated Abs (right). SE short exposure, LE long exposure. Source data are available online for this figure.

the UBE2F-SAG axis. We also examined the mTorc1 activity in *Sag* knockout MEF cells (Tan et al, 2015), and found that like *Ube2f* knockout, *Sag* knockout impaired the mTorc1 activity at both basal and serum-induced levels (Fig. EV3A).

The mTOR is a central hub of the signal transductions, and precisely coordinates the cell growth by integrating many environmental cues, such as growth factors, glucose, and amino acids (Kim et al, 2002). We next investigated whether the manipulation of UBE2F and SAG would affect mTORC1 activation triggered by glucose and amino acids. Indeed, knockdown of either UBE2F or SAG blocked full activation of mTORC1 stimulated by both amino acid and glucose (Figs. 3B,C and EV3B–G) in multiple cancer cell lines, implying that the UBE2F-SAG axis affects intrinsic activity of mTORC1. Likewise, overexpression of wild-type UBE2F, but not its enzymatic-dead UBE2F-C116A mutant, was capable of activating S6K1 phosphorylation, suggesting that UBE2F enzymatic activity is necessary to maintain active mTORC1 status under these conditions (Fig. EV3H).

It has been well-established that RAG complex and RHEB GTPase are the most proximal activators of mTORC1 under amino acid stimulated condition (Bar-Peled and Sabatini, 2014). To assess which GTPase is selectively involved in this process, PLC/PRF/5 cells were transfected with constitutively active RHEB (RHEB-Q64L) or co-transfected with constitutively active RAG complex (RagC-S75N and RagB-Q99L). Interestingly, only ectopic expression of RHEB-Q64L, but not RAG complex, rescued mTORC1 inactivation by UBE2F or SAG knockdown upon amino acid stimulation (Fig. 3D,E). Consistent with this, knockdown of UBE2F or SAG did not alter lysosome localization of mTOR (Fig. EV3I,J), implying that the UBE2F-SAG axis regulates the mTORC1 activity in a manner dependent on RHEB.

## The UBE2F-SAG axis promotes RHEB neddylation

We then tested our working hypothesis that RHEB is subjected to neddylation modification by UBE2F-SAG, the neddylation E2-E3 axis. We first determined the potential interactions between RHEB and the component of UBE2F-SAG axis, along with the components of the UBE2M-RBX1 axis, as the negative control. While both UBE2F and UBE2M E2 failed to bind to RHEB, ectopically expressed SAG E3, but not RBX1 E3, directly interacts with endogenous RHEB (Fig. EV4A,B). Importantly, under unstressed physiological condition, endogenous SAG bound to endogenous RHEB, but neither SAG nor RHEB binds to UBE2F or mTOR in these liver cancer cell lines (Fig. 4A,B). Although SAG is a dual E3 ligase for both neddylation and CRL5-driven ubiquitylation, the fact that CUL-5 knockdown had minimal, if any, effect on mTORC1 activity highly suggested that SAG acts as a neddylation

E3 upon binding with RHEB. The in vitro binding assay using purified proteins showed that SAG directly interacts with RHEB in the absence of CUL-5 (Fig. 4C), further supporting that (1) SAG directly binds to RHEB, and (2) the SAG-RHEB binding is CUL-5 independent.

We next investigated whether RHEB can indeed be neddylated. A plasmid encoding FLAG-tagged RHEB was transfected into HEK293 cells, followed by immunoprecipitation using anti-FLAG beads. Notably, NEDD8 modification of ectopically expressed RHEB was detectable as a slower-migrating band, which was removed by the treatment of MLN4924, a specific neddylation inhibitor (Fig. 4D) (Soucy et al, 2009). More importantly, although at a low percentage, as compared to non-neddylated population, endogenous RHEB was indeed neddylated under the physiological conditions, which was blocked by MLN4924 treatment (Figs. 4E,F and EV4C,D).

To further validate RHEB neddylation, we performed an in vivo neddylation assay, and found that cotransfection of His-NEDD8 and HA-RHEB triggered the formation of poly-neddylated RHEB, which was completely blocked by MLN4924, largely blocked by knockdown of either UBE2F or SAG, but not at all by UBE2M knockdown (Fig. 4G). A much weak blockage by TAK243, a UAE inhibitor (Hyer et al, 2018), suggested minor incorporation of NEDD8 into polyneddylated chain of RHEB, catalyzed by UAE (Fig. 4G) (Hjerpe et al, 2012). Thus, it appears that RHEB serves as a substrate of neddylation.

We next performed an in vitro neddylation assay, using purified proteins and found that RHEB poly-neddylation formed only in the presence of NEDD8, NAE E1, UBE2F E2, and SAG E3. Missing of any one of these components abrogated RHEB neddylation. Furthermore, RHEB-K169R mutant, a neddylation dead mutant (defined in Fig. 5) was not neddylated by the UBE2F-SAG axis (Fig. 4H). We also assessed the potential interplay between neddylation and ubiquitylation of RHEB, and found that RHEB-K169R mutant was also resistant to polyubiquitylation (Appendix Fig. S2A), suggesting that the K169 is subjected to both neddylation and ubiquitylation. We then determined whether UBE2F knockdown affects RHEB ubiquitylation, and found that while UBE2F knockdown inhibited RHEB neddylation, it has no effect on RHEB ubiquitylation (Appendix Fig. S2B). Taken together, RHEB is indeed a bona fide neddylation substrate by the UBE2F-SAG E2-E3 pair.

## Neddylation K169R mutant loses RHEB activity with altered lysosome localization

RHEB protein contains 14 lysine residues. To define the neddylation site on RHEB, a total of 14 single K→R mutants were

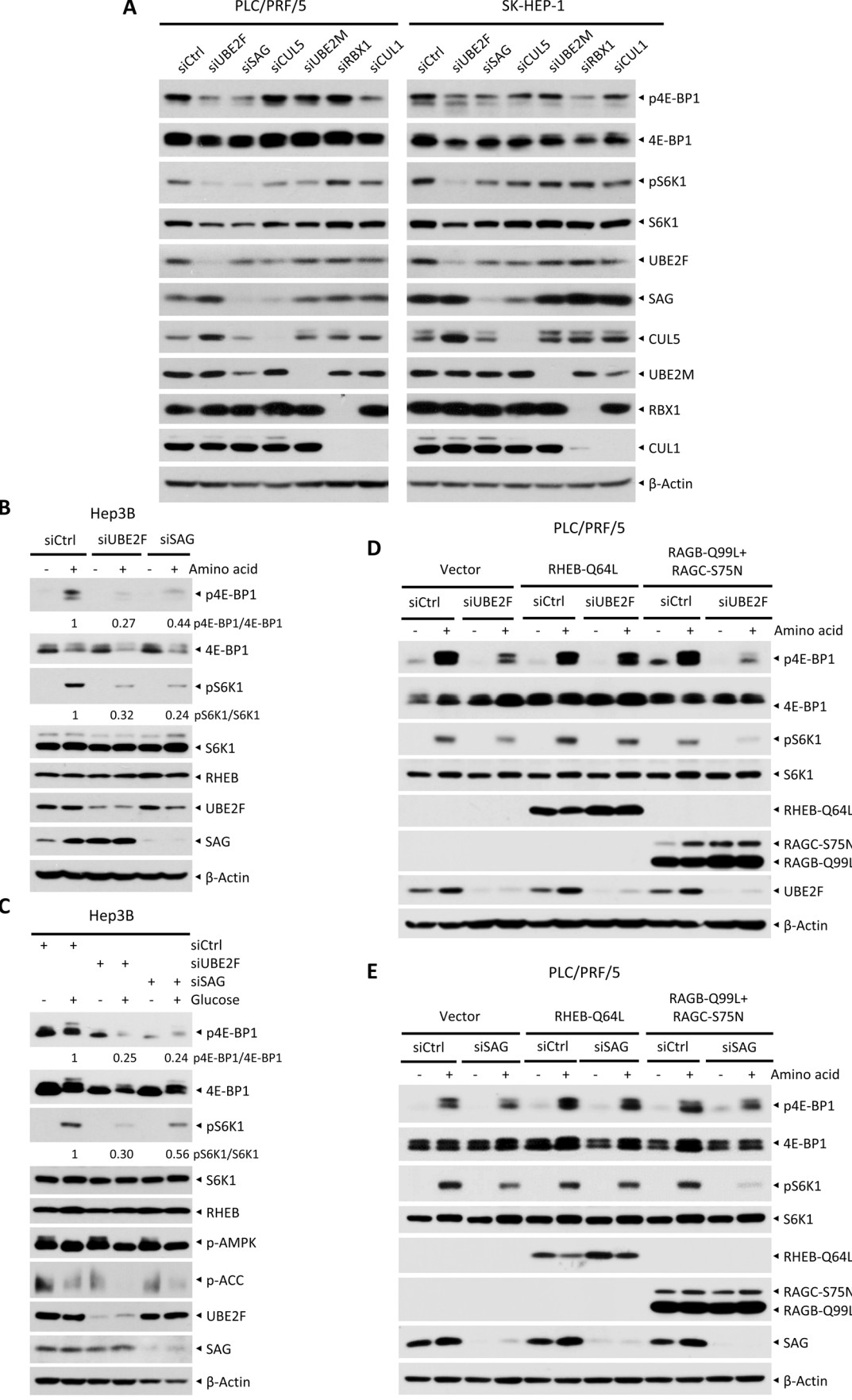

◄ **Figure 3.   The UBE2F-SAG axis regulates mTORC1 activity in a RHEB-dependent manner.**

(A) PLC/PRF/5 and SK-HEP-1 cells were transfected with indicated siRNAs for 72 h, followed by IB analysis. (B) Cells transfected with indicated siRNAs were deprived of amino acids for 50 min, and then re-stimulated with amino acids for 15 min, followed by IB analysis with indicated Abs. (C) Cells transfected with indicated siRNAs were starved of glucose for 6 h, and then re-stimulated with glucose for 20 min, followed by IB analysis. (D, E) PLC/PRF/5 cells were simultaneously co-transfected with indicated siRNAs and HA-tagged RHEB-Q64L or RAGB-Q99L and RAGC-S75N for 72 h, and then starved of amino acids for 50 min, followed by re-supply of amino acids for 15 min before being harvested for IB analysis with indicated Abs. Source data are available online for this figure.

generated. The in vivo neddylation assay showed that while few other mutants were partially abrogated for neddylation, RHEB-K169R mutant was completely abrogated (Fig. 5A), indicating that K169 is the predominant site for RHEB neddylation. Importantly, the RHEB-K169 is an evolutionarily conserved site among few species (Fig. EV5A).

Neddylation modification of a protein normally alters its activity and subcellular localization, but not for targeted degradation (Zhao et al, 2014). We first assessed RHEB protein half-life and found that neddylation-dead mutant RHEB-K169R had similar protein half-life as wild type (wt) RHEB, indicating that the abrogation of neddylation modification did not affect the RHEB stability (Fig. EV5B,C). We next investigated whether RHEB neddylation affects its activity, using p4E-BP1/pS6K1 levels as the readout, given RHEB is a positive regulator of mTORC1 activity (Long et al, 2005). Indeed, the siRNA-based RHEB knockdown inactivated mTORC1 activity, as reflected by reduced p4E-BP1 and pS6K1 levels. While ectopically expressed wt RHEB rescued mTORC1 inactivation, the RHEB-K169R mutant largely lost this rescuing activity (Figs. 5B and EV5D), indicating that RHEB neddylation at K169 is required for its positive regulation of mTORC1 activity.

RHEB consists of 184 amino acids with the first 169 amino acids constituting the GTPase domain, and a CAAX motif for farnesylation at the C-terminus (Fig. EV5E). The CAAX motif is essential for RHEB lysosome localization with the last 15 amino acids acting as the lysosome targeting signal (Angarola and Ferguson, 2019; Sancak et al, 2010). It is well-established that lysosome localization and GTP-binding are required for RHEB to fully activate mTORC1 (Angarola and Ferguson, 2020; Inoki et al, 2003a). Interestingly, the K169 residue is situated in the junction of these two functional domains (Angarola and Ferguson, 2020), suggesting that neddylation modification on K169 may impact both lysosome localization and GTP-binding activity.

We first determined whether K169R mutation alters the lysosome localization of RHEB, and found that indeed, unlike wild type RHEB, RHEB-K169R mutant was not enriched within the lysosome fractions (Fig. 5C and EV5F, top panels). Moreover, compared to wild type RHEB, RHEB-K169R mutant displayed a dispersed subcellular distribution and diminished co-localization with lysosomal marker LAMP2 (Figs. 5C, EV5F, bottom panels and EV5G). Furthermore, overexpression of NEDD8 facilitated lysosome enrichment of WT-RHEB, but not RHEB-K169R mutant (Fig. EV5H,I), and consequent activation of mTORC1 by WT-RHEB, but not by RHEB-K169R mutant (Fig. EV5J), indicating neddylation is indispensable for RHEB subcellular location.

Given that RHEB K169 site is spatially proximal to its C-terminal farnesylation site (C181) for membrane targeting, we tested our hypothesis that neddylation may increase RHEB farnesylation, as a mechanism for enhanced lysosomal localization. We, therefore, generated the RHEB-C181S, a well-known RHEB un-farnesylated

mutant (Hanker et al, 2010). Using the immunofluorescence staining, we found that both K169R and C181S mutants displayed a similar dispersed subcellular distribution (Appendix Fig. S3), highly suggesting that neddylation may influence RHEB farnesylation. We further used immunoprecipitation assay to examine the farnesylation levels of WT-RHEB and RHEB-K169R mutant. The results showed that wild-type RHEB is farnesylated, which is inhibited by FTI-277, a farnesylation inhibitor, whereas the farnesylation level of RHEB-K169R mutant is significantly reduced (Fig. 5D). Thus, it appears that RHEB neddylation promotes its lysosome localization by enhancing its farnesylation.

We next assessed the GTP-binding capability of WT-RHEB, RHEB-K169R, or RHEB-Q64L, using an antibody specifically against RHEB$^{GTP}$ form. Strikingly, while both wt and RHEB-Q64L are in a GTP-binding conformation with RHEB-Q64L being more robust, as expected, the RHEB-K169R mutant remarkably lost this active conformation (Fig. 5E), indicating that this neddylation dead mutant is in an inactive status. The molecular dynamics (MD) simulation analysis revealed that the GTP-binding loop of unmodified RHEB has a trend to swing out, while neddylated RHEB preserves a stable conformation in the GTP-binding loop which forms a hydrogen bond between RHEB and GTP (Fig. 5F), highly suggesting that neddylation of RHEB may stabilize its binding with GTP.

To further investigate the impact of the K169 neddylation on GTP binding, we introduced an additional K169R mutation on the RHEB-Q64L construct, known for its high GTP-binding activity, to generate the RHEB double mutant RHEB-Q64L-K169R, designated as RHEB-DM in this study. In comparison to the constitutively active RHEB-Q64L form, RHEB-DM displayed a reduced mTORC1-activating activity (Figs. 5G and EV5K).

To elucidate the functional consequences of neddylation modification, we assessed whether the RHEB-K169R mutant could effectively rescue the mTORC1 activity inhibited by knockdown of UBE2F or SAG. While RHEB-Q64L could largely rescue the mTORC1 inactivation, RHEB-DM mutant had much reduced activity (Figs. 5H and EV5L). In the same vein, WT-RHEB effectively rescued mTORC1 inactivation by knockdown of UBE2F or SAG, whereas RHEB-K169R mutant largely failed to do so (Fig. EV5M,N). Taken together, neddylation modification appears to be required for proper lysosomal localization of RHEB and active GTP-bound conformation. Neddylation abrogation, as shown in neddylation-dead mutant RHEB-K169R, ablates both capacities, leading to impairment for mTORC1 activation.

## RHEB-K169R mutant fails to rescue altered growth phenotypes induced by knockdown of UBE2F or SAG

We next investigated the biological consequence of RHEB-K169R mutation in liver cancer cells. Ectopic expression of WT-RHEB, but

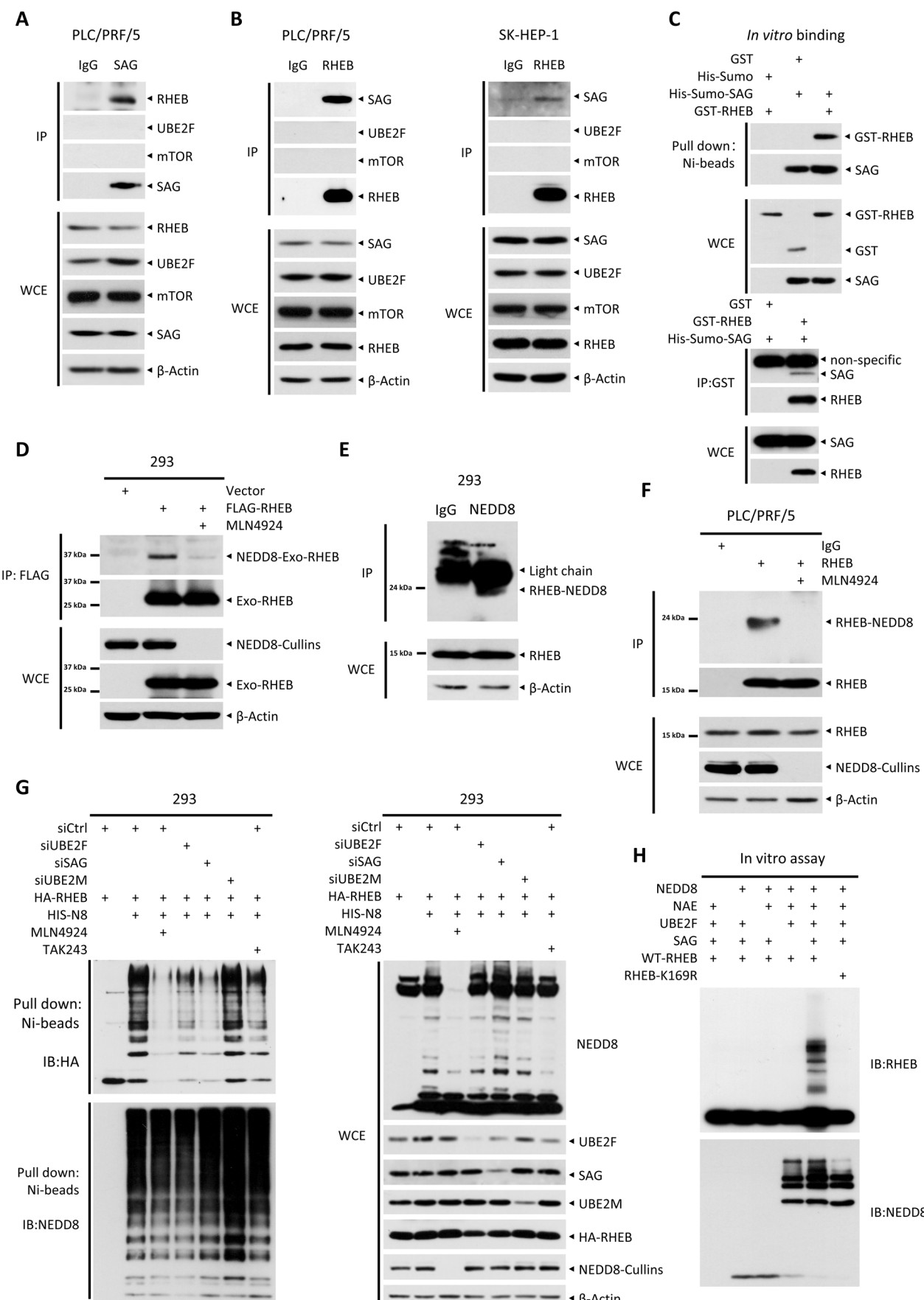

Figure 4.    The UBE2F-SAG axis promotes RHEB neddylation.

(A, B) Cells were harvested and then subjected to IP with anti-SAG Ab (A) or anti-RHEB Ab (B), along with normal IgG, followed by IB analysis with indicated Abs. (C) A total of 100 ng purified GST-tagged proteins and 50 ng His-SUMO-tagged proteins were mixed in binding buffer and subjected to pull-down using Ni beads (top) or GST beads (bottom), followed by IB analysis with the specified antibodies. (D) HEK293 cells were transfected with plasmids expressing FLAG-RHEB and then treated with 1 μM MLN4924 or DMSO as a control for 24 h before being harvested. The cell lysates were then incubated overnight with FLAG beads and subjected to IB using an anti-NEDD8 Ab to detect exogenous RHEB neddylation. (E) HEK293 cell lysates were subjected to IP using an anti-NEDD8 Ab, with normal IgG as a control, followed by IB with an anti-RHEB Ab. (F) PLC/PRF/5 cells were treated with 1 μM MLN4924 or DMSO as a control for 24 h before being harvested for IP using an anti-RHEB Ab, along with normal IgG as a control. The samples were then analyzed by IB with an anti-NEDD8 Ab to detect endogenous RHEB neddylation. (G) HEK293 cells were transfected with the indicated siRNAs for 24 h, followed by transfection with 2 μg HA-RHEB and 1.5 μg HIS-NEDD8 (HIS-N8) for an additional 48 h. The cells were treated with 0.2 μM TAK243 or 1 μM MLN4924 for 12 h, and then lysed under denatured condition. His-tagged neddylated proteins were pulled down by Ni-NTA beads, and then subjected to IB analysis with indicated Abs. (H) 100 ng of purified WT-RHEB or RHEB-K169R proteins were added to a reaction mixture containing 200 μM ATP, 0.3 μM NEDD8, 0.025 μM NAE, 0.8 μM UBE2F, and 200 ng of SAG proteins, as indicated. The mixture was continuously mixed for 90 min at 37 °C before being subjected to IB analysis. Source data are available online for this figure.

not RHEB-K169R mutant promoted cell growth, and growth suppression induced by UBE2F knockdown can be largely rescued by WT-RHEB, but not K169R mutant (Figs. 6A and EV6A). Like UBE2F knockdown, SAG knockdown also suppressed growth, which was also largely rescued by WT-RHEB, but by not K169R mutant (Figs. 6A and EV6A). The same results were seen in clonogenic survival assay, in which suppression induced by knockdown of UBE2F or SAG can be largely rescued by WT-RHEB, but not by K169R mutant (Fig. 6B,C). Consistently, overexpression of UBE2F or SAG activated mTORC1 activity and stimulated cell growth. Importantly, this effect can be rescued by rapamycin treatment (Figs. 6D–F and EV6B–D), suggesting that the growth-promoting effects of UBE2F and SAG are mediated through mTORC1 activation. Moreover, the cell size reduction induced by RHEB knockdown was rescued by RHEB-Q64L, but not by RHEB-DM (Fig. 6G,H).

We further pursued autophagy phenotype and found that again autophagy induced by knockdown of either UBE2F or SAG can be largely rescued by WT-RHEB, but not by K169R mutant (Fig. EV6E–H). Similarly, LC3-II accumulation and p62 degradation induced by knockdown of either UBE2F or SAG can be largely rescued by RHEB-Q64L, while RHEB-DM was less effective (Figs. 6I–L and EV6I,J). Taken together, it appears that neddylation modification regulates both biochemical activity and biological functions of RHEB.

## Ube2f deletion attenuates liver steatosis and tumorigenesis in vivo

Finally, we investigated the role of Ube2f in liver tumorigenesis, using a well-established mouse liver tumor model, driven by liver selective Pten deletion induced by Alb-Cre. We reasoned that Pten loss activates the mTOR signals to induce liver tumorigenesis, whereas Ube2f deletion would inactivate mTORC1 which should block tumorigenesis, if our in vitro cell culture work can be extended to an in vivo physiological setting. In the Alb-Cre;pTen^fl/fl model, the development of liver steatosis occurs at 10-12 weeks, followed by appearance of hepatocellular carcinoma (HCC) at ~12-16 months (Galicia et al, 2010). We first examined liver steatosis by harvesting the livers from male paired litter-mates with the genotypes of Alb-Cre;pTen^fl/fl;Ube2f^+/+ and Alb-Cre;pTen^fl/fl;Ube2f^fl/fl for Oil-red-O staining. As expected, the Pten loss triggered liver steatosis with increased severity as disease progression with the time (3 vs. 6 months). Remarkably, Ube2f deletion significantly delayed/inhibited the liver steatosis (Fig. 7A–C). Mechanistically,

Ube2f deletion reduced both mRNA and protein levels of Srebp1c, a known mTorc1 target gene (Fig. EV7A–D) (Shimano and Sato, 2017), indicating mTorc1 inactivation.

We then evaluated Ube2f effect on liver tumorigenesis. Notably, Ube2f deletion inhibited liver tumorigenesis, as evidenced by reduced number and size of liver tumors as well as liver weight (Figs. 7D and EV7E,F), along with reduced staining of p-4Ebp1 and p-S6 (Figs. 7E and EV7G,H), again indicating mTorc1 inactivation. Furthermore, regardless of Ube2f, compared to normal tissues, tumor tissues had the higher levels of pS6. However, even higher levels of pS6 were seen in liver tumors with wild type Ube2f than that with Ube2f deletion (Fig. 7F), further supporting the oncogenic role of Ube2f via activating mTorc1.

We further observed that Ube2f deletion triggered remarkable hepatic cystogenesis in this Pten-null liver cancer model (Fig. 7G,H). The formation of liver cysts is attributed to the defects in the remodeling of the ductal plate during biliary tract development, known as ductal plate malformation (Wills et al, 2014). Indeed, we found that the cysts in the Ube2f-Pten double null liver were stained positively for CK-19 (Fig. EV7I), a well-established marker for biliary cells (Carpino et al, 2018). Notably, Sox9, a well-known transcription factor in governing bile duct differentiation (Antoniou et al, 2009) was significantly down-regulated upon Ube2f deletion (Figs. 7I and EV7J). While tumor tissues had increased levels of Sox9 protein and mRNA, Ube2f deletion reduced both (Figs. 7I and EV7J,K).

Following this lead, we performed siRNA based study in liver cancer cells, and found that PTEN knockdown indeed activated mTORC1 activity with increased level of pS6K1 and pS6. Importantly, simultaneous knockdown of UBE2F or SAG with PTEN inactivated mTORC1 activity and downregulated SOX9 expression (Figs. 7J and EV7L). Consistently, overexpression of UBE2F or SAG upregulated SOX9 protein level, which was largely rescued by rapamycin treatment (Fig. EV7M). The observation of a positive correlation between SOX9 level and mTORC1 activity strongly suggests that Sox9 is an mTORC1 target gene, consistent with the observation that Sox9 is subjected to mTORC1 regulation (Iezaki et al, 2018).

## High UBE2F levels and mTORC1 activity in liver cancer tissues correlate with poor patient survival

Finally, we determined whether UBE2F regulation of the mTORC1 signal can be extended to human liver cancer tissues.

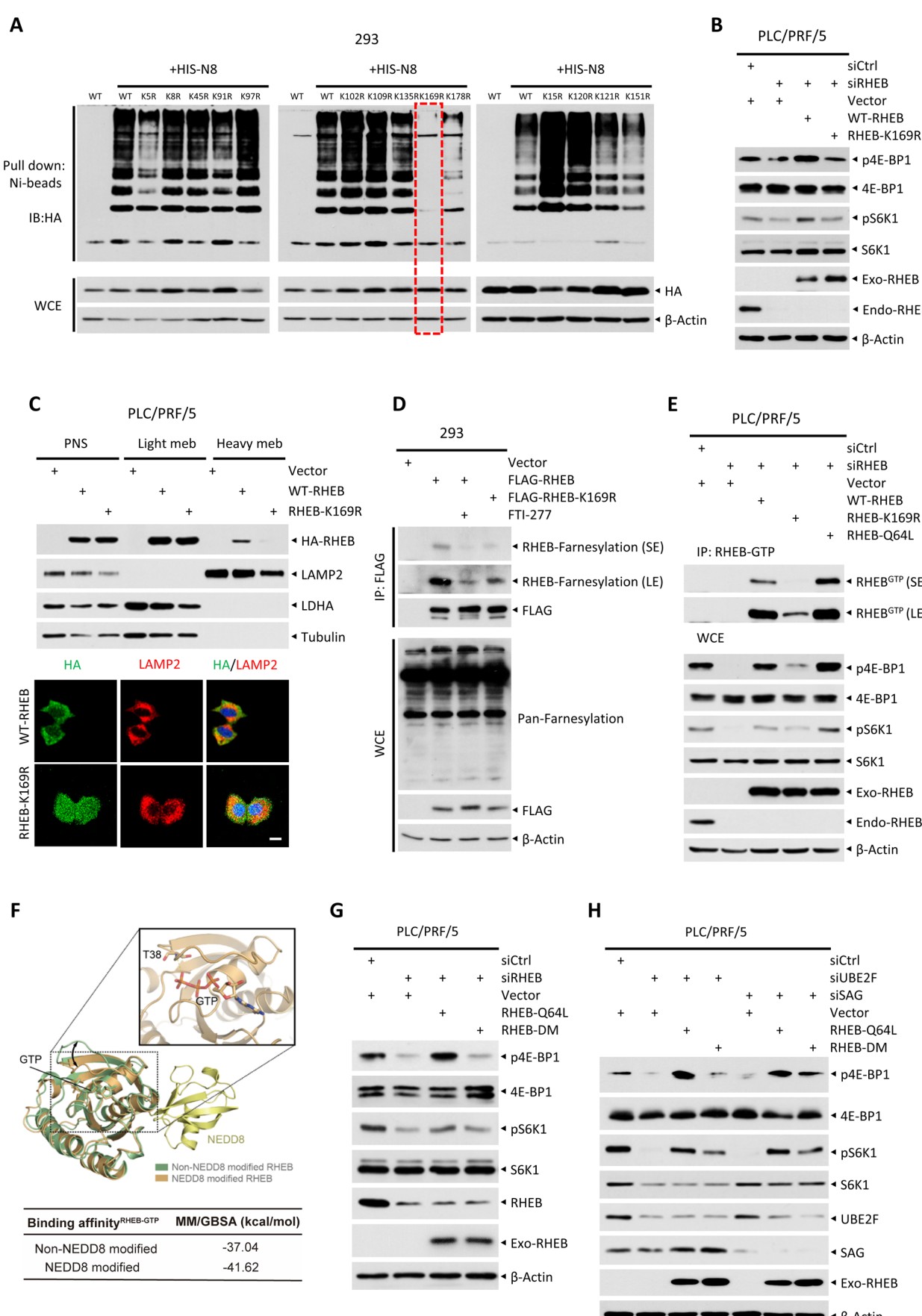

**Figure 5.   Neddylation K169R mutant loses RHEB activity with altered lysosome localization.**

(A) HEK293 cells were transfected with 2 μg HA-tagged mutants of RHEB, along with 1.5 μg HIS-N8 for 48 h. Cells were harvested, and then lysed in denatured lysis buffer. Neddylated proteins were pulled down using Ni-NTA beads, followed by IB analysis using an anti-HA Ab. The red box indicates that the RHEB-K169R mutant cannot be neddylated. (B) Cells were simultaneously transfected with the indicated siRNAs and 3 μg of either WT-RHEB or RHEB-K169R plasmids for 72 h, and then subjected to IB analysis. (C) PLC/PRF/5 cells were transfected with 15 μg of HA-WT-RHEB or HA-RHEB-K169R mutant per 15 cm dish. Cells were then subjected to IB analysis after the isolation of heavy membrane and light membrane fractions (top) or immunofluorescent staining of HA (green) and LAMP2 (red) (bottom). PNS: post-nuclear supernatants. Scale bar, 10 μm. (D) HEK293 cells were transfected with the indicated plasmids and subsequently treated with 10 μM FTI-277 or vehicle control overnight. Cell lysates were incubated with FLAG beads, followed by IB analysis with an anti-pan-farnesylation Ab to detect farnesylation of WT-RHEB and RHEB-K169R. (E) Cells were simultaneously transfected with the indicated siRNAs and 10 μg of FLAG-tagged plasmids per 10 cm dish for 72 h, and then subjected to IP with anti-RHEB$^{GTP}$ Ab, followed by IB with an anti-FLAG Ab to detect the GTP binding status of exogenous RHEB. (F) Structural superimposition of unmodified or NEDD8-modified RHEB bound with GTP from AMBER MD simulation. RHEB with and without NEDD8 modification are shown in green and orange, respectively. NEDD8 is shown in yellow. Conformational change of the GTP-binding loop is indicated by the black double-arrow line. Enlargement of the GTP-binding site in NEDD8-modified RHEB shows the hydrogen bond formed between the main chain of T38 residue and the GTP (top). The relative binding energy of non-NEDD8 modified and NEDD8 modified RHEB with GTP was shown (bottom). (G, H) Cells were transfected with indicated siRNAs and plasmids for 72 h, followed by IB analysis with indicated Abs. RHEB-DM: RHEB-Q64L-K169R. SE short exposure, LE long exposure. Source data are available online for this figure.

Immunohistochemistry staining of liver tumor tissue microarray (containing 81 individual tumor tissues) showed that the high levels of UBE2F and mTORC1 activity, as reflected by pS6 staining, were indeed positively correlated (Fig. 7K,L). Importantly, higher levels of both UBE2F and pS6 were associated with worse patient survival (Fig. 7M), suggesting that UBE2F cooperates with mTORC1 to promote liver tumorigenesis.

## Discussion

In this study, we demonstrated biochemically that the UBE2F-SAG E2-E3 pair neddylates RHEB to enhance RHEB lysosome localization and GTPase activity, leading to mTORC1 activation. Biologically, UBE2F knockdown in liver cancer cells inactivates mTORC1 to suppress cell growth, reduce cell size and induce autophagy, whereas liver-specific *Ube2f* deletion significantly inhibits liver steatosis and tumorigenesis induced by *Pten* loss.

Our previous studies showed that UBE2F knockdown suppressed growth and survival of lung cancer cells via inducing NOXA accumulation to induce apoptosis (Zhou et al, 2017). Our most recent study showed that UBE2F knockdown inhibited growth and survival of pancreatic cancer cells in vitro, and suppressed pancreatic tumorigenesis induced by *Kras*$^{G12D}$ in vivo via inducing accumulation of DIRAS2, a RAS inhibitor (Kontani et al, 2002) to block the MAPK/c-Myc signals (Chang et al, 2024). Other studies have shown that treatment of cancer cells with irradiation (Lisha et al, 2021) or platinum (Zhou et al, 2020) induced UBE2F expression to ubiquitylate and degrade NOXA, leading to resistance to radiotherapy or platinum-based chemotherapy, respectively. Consistently, we found that a small molecule neddylation inhibitor of UBE2F-Cullin-5 effectively sensitized lung cancer cells to radiation (Xu et al, 2022b). Whether and how UBE2F regulates growth of liver cancer cells and liver tumorigenesis is previously unknown.

To this end, we first analyzed the TCGA liver cancer database, and found that compared to normal liver tissue, UBE2F is overexpressed in liver cancer tissues, which is positively correlated with the poor survival of patients. This finding was further confirmed by immune-histochemical staining using liver tumor tissue microarray samples. To determine the biological significance, we used few lines of liver cancer cells and showed that UBE2F knockdown suppresses the growth and survival of liver cancer cells,

indicating its growth-essential role. Further characterization of nature of growth inhibition revealed that unlike lung cancer cells (Zhou et al, 2017), UBE2F knockdown in liver cancer cells did not induce apoptosis, rather reduced the cell size and induced autophagy, the signs of mTORC1 inactivation. The follow-up biochemical studies confirmed mTORC1 inactivation, as evidenced by reduced levels of p4E-BP1 and pS6K in both liver cancer cells and MEF cells upon UBE2F knockdown or knockout, respectively. Interestingly, the mTOCR1 inactivation occurs under both basal and stressed conditions such as with stimulation by glucose and amino acids. Furthermore, the mTORC1 inactivation is rather specific for knockdown of the UBE2F-SAG pair, but not the UBE2M-RBX1 pair.

How does UBE2F knockdown selectively trigger mTORC1 inactivation? We first excluded the possible accumulation of mTORC1 negative regulators, such as DEPTOR and TSC1/2, known as the substrates of CRLs (Hu et al, 2008; Luo et al, 2012; Zhao et al, 2011). We next excluded the possibility that UBE2F may directly form a complex with mTORC1. We then focused on mTORC1 positive regulator, particularly RHEB. It is well-established that mTORC1 is activated by two constitutively active forms of GTPase: RHEB-Q64L, and RAGB-Q99L/RAGC-S75N (Bar-Peled et al, 2013). Our rescue experiments showed that RHEB-Q64L, but not RAGB-Q99L/RAGC-S75N blocks the mTORC1 inactivation induced by knockdown of UBE2F or SAG, respectively. Thus, RHEB is the key regulator that mediates the effect of UBE2F/SAG knockdown.

We then tested our working hypothesis that RHEB is a neddylation substrate of the UBE2F/SAG, but not UBE2M/RBX1 pair, which is indeed the case, supported by the following lines of evidence: (1) neddylation E3 SAG directly binds to RHEB in both in vitro and in vivo pull-down assays; (2) endogenous RHEB-NEDD8, although at very low level, is detectable in cells, and NEDD8 attachment to both endogenous and exogenous RHEB is completely blocked by MLN4924 treatment; (3) the in vivo neddylation assay showed that RHEB is neddylated by the UBE2F-SAG pairs, but not UBE2M-RBX1 pairs, which is abrogated by neddylation E1 inhibitor, MLN4924, but not ubiquitylation E1 inhibitor TAK243; (4) the UBE2F-SAG pair directly neddylates RHEB in an in vitro neddylation assay; and (5) RHEB is neddylated on the K169 site. A previous study showed that RHEB is subjected to ubiquitylation regulation, which governs the nucleotide-bound status of Rheb. Specifically, RNF152, a lysosome-anchored E3

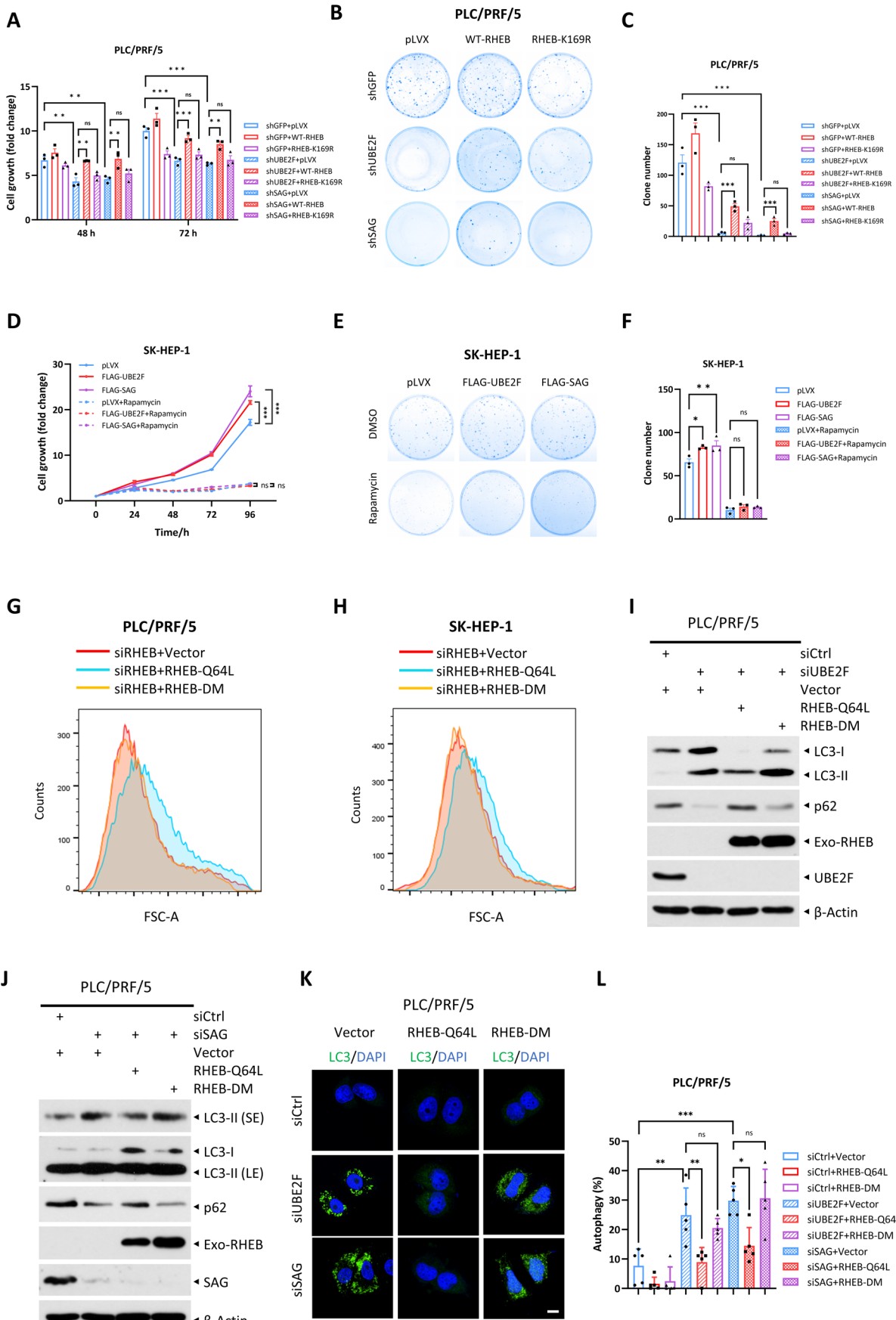

◄

**Figure 6. RHEB-K169R mutant fails to rescue altered growth phenotypes induced by knockdown of UBE2F or SAG.**

(A–C) PLC/PRF/5 cells were simultaneously infected with lentivirus-based shRNA targeting UBE2F or SAG, along with lentivirus expressing WT-RHEB or RHEB-K169R. After puromycin selection, stable cell lines were subjected to cell growth assay (A) and clonogenic survival assay (B, C). Representative images of colony formation are shown (B), and colony numbers are plotted (C). Data were shown as mean ± SEM from three independent experiments and analyzed by two-way ANOVA (A) or one-way ANOVA (C). The $P$ values for the comparisons were as follows: (A) shGFP+pLVX vs. shUBE2F+pLVX ($P = 0.0016$ for 48 h and 5.10E-6 for 72 h); shGFP+pLVX vs. shSAG+pLVX ($P = 0.0066$ for 48 h and 7.37E-7 for 72 h); shUBE2F+pLVX vs. shUBE2F+WT-RHEB ($P = 0.0022$ for 48 h and 6.05E-4 for 72 h); shUBE2F+pLVX vs. shUBE2F+RHEB-K169R ($P = 0.9233$ for 48 h and 0.9161 for 72 h); shSAG+pLVX vs. shSAG+WT-RHEB ($P = 0.0026$ for 48 h and 0.0037 for 72 h); shSAG+pLVX vs. shSAG+RHEB-K169R ($P = 0.9482$ for 48 h and 0.9938 for 72 h). (C) shGFP+pLVX vs. shUBE2F+pLVX ($P = 1.04E-7$); shGFP+pLVX vs. shSAG+pLVX ($P = 6.64E-8$); shUBE2F+pLVX vs. shUBE2F+WT-RHEB ($P = 9.29E-4$); shUBE2F+pLVX vs. shUBE2F+RHEB-K169R ($P = 0.0767$); shSAG+pLVX vs. shSAG+WT-RHEB ($P = 1.45E-4$); shSAG+pLVX vs. shSAG+RHEB-K169R ($P = 0.3987$). **$P < 0.01$; ***$P < 0.001$; ns no significant. (D–F) SK-HEP-1 cells stably overexpressing FLAG-tagged UBE2F or SAG via a lentivirus-based approach were treated with or without 1 µM rapamycin, and then subjected to cell growth assay (D) and clonogenic survival assay (E, F). Cells were cultured in 2.5% FBS for cell growth assay, representative images of colonies are shown (E), and colony numbers are plotted (F). Data were shown as mean ± SEM from three independent experiments and analyzed by two-way ANOVA (D) or one-way ANOVA (F), respectively. The $P$ values for the comparisons were as follows: (D) pLVX vs. FLAG-UBE2F ($P = 2E-11$); pLVX vs. FLAG-SAG ($P = 1.99E-11$); pLVX+Rapamycin vs. FLAG-UBE2F+Rapamycin ($P = 0.9843$); pLVX+Rapamycin vs. FLAG-SAG +Rapamycin ($P = 0.8243$). (F) pLVX vs. FLAG-UBE2F ($P = 0.0196$); pLVX vs. FLAG-SAG ($P = 0.0079$); pLVX+Rapamycin vs. FLAG-UBE2F+Rapamycin ($P = 0.9334$); pLVX+Rapamycin vs. FLAG-SAG+Rapamycin ($P = 0.9680$). *$P < 0.05$; **$P < 0.01$; ***$P < 0.001$; ns: no significant. (G, H) Cells were transfected with the indicated siRNA and plasmids for 72 h. The cells were then trypsinized and resuspended in PBS before being subjected to cell size measurement. (I–L) PLC/PRF/5 cells were transfected with indicated siRNAs and plasmids for 72 h. Cells were then harvested for IB analysis with indicated Abs (I, J), or subjected to immunofluorescent co-staining of LC3 (green) and DAPI (blue) (K), and the statistical analysis of the percentages of cells exhibiting LC3-positive puncta are shown (L). Data were shown as mean ± SD from five random fields and analyzed by one-way ANOVA. The $P$ values were as follows: siCtrl+Vector vs. siUBE2F+Vector ($P = 0.0025$); siCtrl+Vector vs. siSAG+Vector ($P = 5.85E-5$); siUBE2F+Vector vs. siUBE2F+RHEB-Q64L ($P = 0.0061$); siUBE2F+Vector vs. siUBE2F+RHEB-DM ($P = 0.9672$); siSAG+Vector vs. siSAG+RHEB-Q64L ($P = 0.0099$); siSAG+Vector vs. siSAG+RHEB-DM ($P = 0.9999$). *$P < 0.05$; **$P < 0.01$; ns: no significant. Scale bar, 10 µm. SE short exposure, LE long exposure. Source data are available online for this figure.

ligase, catalyzes Rheb ubiquitylation at K8 site, which promotes its binding to the TSC complex and facilitates GTP hydrolysis, eventually leading to mTORC1 inactivation (Deng et al, 2019). Thus, it appears that RHEB neddylation and ubiquitylation has an opposite effect on RHEB to regulate mTORC1 activity.

It is well-established that RHEB is a small GTPase that directly activates mTORC1 activity (Li et al, 2004), but is negatively regulated by tumor suppressors TSC1/TSC2. Loss of TSC1/2 function leads to tumor formation in various organs, an autosomal inherited disorder known as Tuberous sclerosis complex (TSC) (Narayanan, 2003). However, mechanistically how RHEB stability or activity is precisely regulated remains elusive. It was reported that in the heart, Rheb is an Atg6 downstream target gene, required for growth of cardiac myocytes induced by phenylephrine and IGF1. Likewise, in Atf6 cKO mice, ectopic expression of Rheb effectively restored the loss of mTorc1 signaling, cardiac function, and Atf6 target gene expression (Blackwood et al, 2019). While a recent study reported that energy status regulated RHEB levels (Li et al, 2024), our results showed that RHEB levels remained stable under the starvation-refeeding of glucose, amino acid, and serum (Figs. 2D, 3B,C and EV3A). This might be due to cell line-dependent discrepancy. Unlike RAGs, the RHEB GTPase lacks a guanine nucleotide exchange factor (GEF), leading to a higher GTP-bound status under normal culture conditions (Emmanuel et al, 2017). This elevated GTP binding status may render RHEB more sensitive to cellular stress. Indeed, in ovarian cancer cells, the GTPase RGS10 accelerated the hydrolysis of GTP bound to RHEB, thereby inactivating RHEB and mTORC1 activity (Altman et al, 2015). AG2037, an inhibitor of the purine biosynthetic enzyme GARFT (glycinamide ribonucleotide formyltransferase), reduced intracellular guanine nucleotides, subsequently inhibited GTP-bound RHEB, leading to mTORC1 inactivation (Emmanuel et al, 2017).

Similar to the RAS superfamily, the C-terminal of RHEB contains a CAAX motif an anchor for farnesylation. However, due to the lack of second membrane-targeting signal (such as the

polybasic region found in Kras, or palmitoylation of upstream cysteines seen in Hras), RHEB has a minimal membrane targeting capacity, which distinguishes it from other members of the RAS superfamily (Angarola and Ferguson, 2019; Buerger et al, 2006). Mutants of RHEB that lack the CAAX motif are unable to activate mTORC1 signaling in cells, underscoring the critical role of farnesylation and the lysosomal localization of RHEB (Buerger et al, 2006; Li et al, 2004; Tee et al, 2003; Zheng et al, 2010). A previous study showed that MCRS1 (microsphere protein 1) maintains RHEB at lysosomal surfaces and connects RHEB to mTOR in an amino acid-dependent manner. Depletion of MCRS1 leads to the accumulation of the GDP-bound form of RHEB, causing its displacement from the lysosomal platform to endocytic recycling vesicles, ultimately resulting in mTORC1 inactivation (Fawal et al, 2015).

What is the consequence of RHEB neddylation at the biochemical and biological levels? In this study, we observed that RHEB is not prominently enriched in lysosomes, as previously reported (Angarola and Ferguson, 2019). The neddylation-deficient mutant RHEB-K169R exhibits a diffuse distribution in both nucleus and cytoplasm, similar to the unfarnesylated mutant RHEB-C181S or after treatment with a farnesyl transferase inhibitor (Hanker et al, 2010), suggesting neddylation is essential for RHEB subcellular location. The fact that RHEB neddylation on K169 enhances its lysosomal localization strongly suggests that it may serve as an additional membrane anchor mechanism. Previous studies have suggested that farnesylation at CAAX motif is crucial for RHEB to effectively bind GTP (Li et al, 2004). Indeed, we found that RHEB neddylation at K169 residue next to CAAX motif promotes its localization to lysosomes by enhancing RHEB farnesylation, and increasing its GTP binding affinity, leading to full activation of RHEB.

It is important to note that lack of K169 neddylation has biological consequence, since mTORC1 inactivation, growth suppression, cell size reduction and autophagy induction in UBE2F-knockdown cells were rescued by wild type RHEB, but

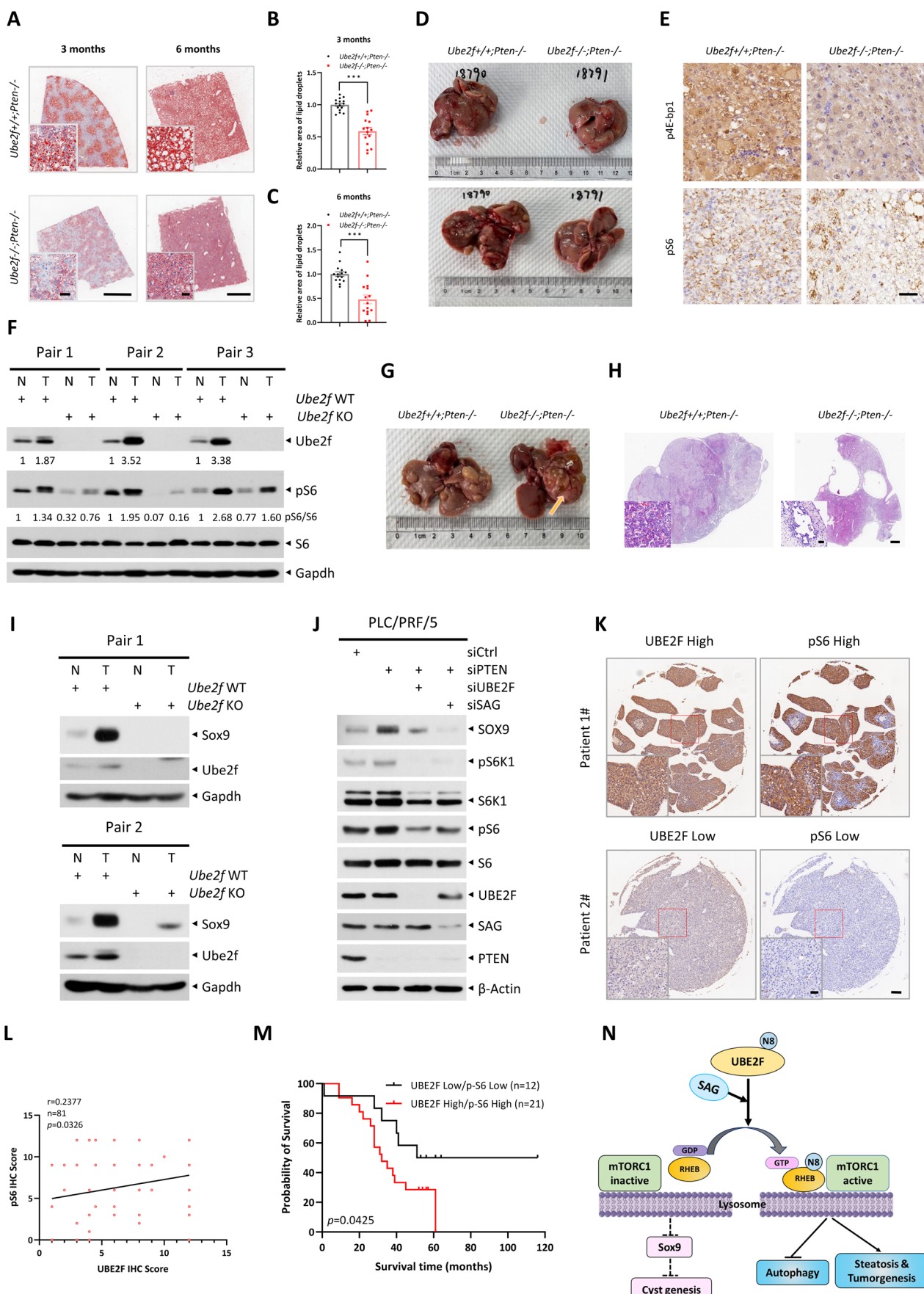

**Figure 7. Ube2f knockout attenuates liver steatosis and tumorigenesis in vivo.**

(A) The livers were harvested from paired littermates with the genotypes of *Alb-Cre;Ube2f$^{+/+}$;Pten$^{fl/fl}$* (*Ube2f + /+;Pten-/-*) and *Alb-Cre;Pten$^{fl/fl}$;Ube2f$^{fl/fl}$* (*Ube2f-/-;Pten-/-*) at 3-month old (left) and 6-month old (right), followed by Oil-red-O staining. Scale bars, 40 μm (inset) and 1 mm, respectively. (B, C) Analysis of lipid droplet area in liver tissues from paired mouse at the ages of 3 months (B) and 6 months (C). The area of lipid droplets was quantified using ImageJ from three pairs of mice at each age. For each tissue, data were obtained from five random fields and normalized to the WT mice data. Results are expressed as mean ± SEM, with statistical significance indicated as ***$P < 0.001$. The P values were: 2.76E-7 for (B); and 2.06E-5 for (C). (D–F) The livers were harvested from paired littermates with the genotypes of *Ube2f + /+;Pten-/-* (*Ube2f* WT) and *Ube2f-/-;Pten-/-* (*Ube2f* KO) at 12-month old (D), followed by immunohistochemical staining (E) or IB analysis (F) with indicated Abs. Scale bar, 40 μm. N normal liver tissues, T tumor tissues. (G) Representative image of hepatic cystogenesis (arrow) in *Ube2f + /+;Pten-/-* and *Ube2f-/-;Pten-/-* mice at 12-month old. (H) H&E staining of liver tissues from (G). Scale bars, 1 mm and 40 μm (inset), respectively. (I) The livers from two paired *Ube2f* WT and KO mice at 12-month old were harvested for IB with indicated Abs. (J) PLC/PRF/5 cells were transfected with indicated siRNAs for 72 h, followed by IB analysis with indicated Abs. (K, L) The liver tumor tissue microarrays containing 81 tumor tissues were stained for UBE2F and pS6 levels. The representative IHC images are shown (K), and the correlation analysis of IHC scores between UBE2F and pS6 is shown (L). Statistical analysis was conducted using simple linear regression. Scale bars, 125 μm and 40 μm (inset), the insets show enlarged images of the red boxes. (M) The Kaplan–Meier survival analysis (using the Mantel–Cox log-rank test) of liver cancer patients with high and low expression of UBE2F and pS6 are shown, both UBE2F and p-S6 IHC scores higher or lower than 6 are referred as UBE2F High/p-S6 High or UBE2F Low/p-S6 Low, respectively. (N) A working model for the UBE2F-SAG axis promoting liver steatosis and tumorigenesis via neddylating RHEB to activate mTORC1. Source data are available online for this figure.

not by RHEB-K169R mutant. In addition, our previous study demonstrated that inhibiting neddylation by MLN4924 suppresses mTORC1, leading to autophagy due to the accumulation of HIF1α, REDD1, and DEPTOR. However, simultaneously knocking down HIF1α, REDD1, or DEPTOR only partially restores mTORC1 activity (Zhao et al, 2012), further suggesting that RHEB neddylation is crucial for the full activation of mTORC1. Collectively, the full activity of RHEB on mTORC1 activation requires its neddylation modification on K169.

Furthermore, using an in vivo liver tumorigenesis model, induced by *Pten* loss, we found that in this mTorc1 activating model, simultaneous *Ube2f* deletion inhibits the mTorc1 signals and suppressed steatosis and tumorigenesis. Thus, Ube2f is an mTorc1 cooperative protein to ensure full mTorc1 activation to promote liver tumorigenesis. Moreover, *Ube2f* appears to play an important role in the fate determination of liver cells during liver tumorigenesis induced by *Pten* deletion, since its deletion inhibits liver tumorigenesis, but promotes liver cytogenesis, in a manner likely regulated by the mTORC1-SOX9 axis. Finally, a positive correlation between high UBE2F levels and more active mTORC1 in human liver tumor tissues highly suggests that UBE2F overexpression seen in liver cancer tissue is not merely a consequence, rather causally related to liver tumorigenesis in humans. Indeed, our previous collaborative study had shown that general neddylation inhibitor MLN4924 inhibited growth of liver cancer cells both in vitro and in vivo by inducing autophagy (Luo et al, 2012). The findings presented in this study further suggested that MLN4924 could potentially enhance the efficacy of anticancer therapies in liver cancers with high UBE2F levels, an interesting subject for future study.

In summary, in this study, we identified and characterized RHEB as the second neddylation substrate, catalyzed by the UBE2F/E2 and SAG/E3 neddylation axis, in addition to well-known cullin-5 as the first substrate. Our study fits the following working model. RHEB, upon neddylation on K169 by the UBE2F-SAG pair of neddylation E2-E3, causes full activation of mTORC1 via increased lysosomal localization and enhanced GTPase activity, leading to growth acceleration and autophagy suppression in cell culture setting, and liver steatosis and tumorigenesis in *Pten*-null in vivo model (Fig. 7N). Thus, targeting the UBE2F-SAG axis may have therapeutic implications for the treatment of non-alcoholic

fatty liver disease and liver cancer with activated mTORC1 signal as a result of *PTEN* loss.

## Methods

**Reagents and tools table**

| Reagent/resource | Reference or source | Identifier or catalog number |
|---|---|---|
| **Experimental models** | | |
| *Ube2f* targeted mouse embryonic stem cells | European Mouse Mutant Cell Repository, eummcr.org (EuMMCR) | Clone ID: HEPD0820_4_A12 |
| *Ube2f$^{flox}$* mice | Chang et al, 2024 | N/A |
| *Pten$^{flox}$* mice | The Jackson Laboratory | Strain# 004597; RRID: IMSR_JAX: 004597 |
| *Alb-cre* mice | The Jackson Laboratory | Strain# 003574; RRID: IMSR_JAX: 003574 |
| PLC/PRF/5 | ATCC | RRID: CVCL_0485 |
| SK-HEP-1 | ATCC | RRID: CVCL_0525 |
| Hep3B | ATCC | RRID: CVCL_0326 |
| HeLa | ATCC | RRID: CVCL_0030 |
| HEK293 | ATCC | RRID: CVCL_0045 |
| **Recombinant DNA** | | |
| FLAG-UBE2F | Zhou et al, 2018b | N/A |
| FLAG-UBE2M | Zhou et al, 2018b | N/A |
| FLAG-SAG | Zhou et al, 2018b | N/A |
| FLAG-CUL1 | Chang et al, 2024 | N/A |
| FLAG-CUL5 | Chang et al, 2024 | N/A |
| HA-RHEB | This paper | N/A |
| FLAG-RHEB | This paper | N/A |
| HA-RHEB-K169R | This paper | N/A |
| FLAG-RHEB-K169R | This paper | N/A |

| Reagent/resource | Reference or source | Identifier or catalog number |
|---|---|---|
| HA-RHEB K5R/K8R/K15R/K45R/ K91R/97R/102R/109R/ 120R/121R/135R/151R /K178R/C181S | This paper | N/A |
| pGEX-4T-1-RHEB | This paper | N/A |
| pGEX-4T-1-RHEB-K169R | This paper | N/A |
| pET-28a-SAG | This paper | N/A |
| **Antibodies** | | |
| Anti-UBE2F antibody | Proteintech | Cat #17056-1-AP; RRID: AB_2210295 |
| Anti-p70 S6 Kinase antibody | Cell Signaling Technology | Cat #2708; RRID: AB_390722 |
| Anti-Phospho-p70 S6K T389 antibody | Cell Signaling Technology | Cat #9234; RRID: AB_2269803 |
| Anti-4E-BP1 antibody | Cell Signaling Technology | Cat #9452; RRID: AB_331692 |
| Anti-Phospho-4E-BP1 T37/46 antibody | Cell Signaling Technology | Cat #2855; RRID: AB_560835 |
| Anti-NEDD8 antibody | Abcam | Cat #ab81264; RRID: AB_1640720 |
| Anti-CUL5 antibody | Abcam | Cat #ab264284; RRID: AB_3073511 |
| Anti-mTOR antibody | Cell Signaling Technology | Cat #2983; RRID: AB_2105622 |
| Anti-RAPTOR antibody | Cell Signaling Technology | Cat #2280; RRID: AB_561245 |
| Anti-Phospho-ACC antibody | Cell Signaling Technology | Cat #3661; RRID: AB_330337 |
| Anti-GAPDH antibody | Cell Signaling Technology | Cat #5174; RRID: AB_10622025 |
| Anti-RHEB antibody | Cell Signaling Technology | Cat #13879; RRID: AB_2721022 |
| Anti-SAG antibody | Proteintech | Cat #11905-1-AP; RRID: AB_10697836 |
| Anti-Farnesylation antibody | Abcam | Cat #ab199481; RRID: AB_299095 |
| Anti-UBE2M antibody | Santa Cruz | Cat #sc-100608; RRID: AB_2211030 |
| Anti-NOXA antibody | Millipore | Cat #OP180; RRID: AB_2268468 |
| Anti-RBX1 antibody | Santa Cruz | Cat #sc-393640; RRID: AB_2722527 |
| Anti-mTOR antibody | Santa Cruz | Cat #sc-517464; RRID: AB_3186240 |
| Anti-FLAG antibody | Sigma-Aldrich | Cat #F1804; RRID: AB_262044 |
| Anti-LAMP2 antibody | Santa Cruz | Cat #sc-18822; RRID: AB_626858 |
| Anti-RHEB antibody | Santa Cruz | Cat #sc-271509; RRID: AB_10659102 |
| Anti-β-Actin antibody | Sigma-Aldrich | Cat #A5441; RRID: AB_476744 |
| Anti-HA antibody | Roche | Cat #11867423001; RRID: AB_390918 |

| Reagent/resource | Reference or source | Identifier or catalog number |
|---|---|---|
| Peroxidase AffiniPure Goat Anti-Rabbit IgG (H + L) | Jackson | Cat #111-035-144 |
| Peroxidase AffiniPure Goat Anti-Mouse IgG (H + L) | Jackson | Cat #115-035-146 |
| Peroxidase AffiniPure Goat Anti-Rat IgG | Jackson | Cat #112-035-143 |
| ALEXA FLUOR 488 Donkey anti-Rabbit IgG (H + L) Highly Cross-Adsorbed Secondary Antibody | Themo | Cat #A21206 |
| ALEXA FLUOR 546 Donkey anti-Mouse IgG (H + L) Highly Cross-Adsorbed Secondary Antibody | Themo | Cat #A10036 |
| **Oligonucleotides** | | |
| siCtrl: 5′-AUU GUA UGC GAU CGC AGA C-3′ | This paper | N/A |
| siUBE2F-1: 5′-CAA AGU GAA AUG CCU GAC CAA-3′ | This paper | N/A |
| siUBE2F-2: 5′-CAU CAA ACG UUA UGC CAG AU-3′ | This paper | N/A |
| siUBE2M: 5′-GGG CUU CUA CAA GAG UGG GAA GU-3′ | This paper | N/A |
| siRBX1: 5′-GAC UUU CCC UGC UGU UAC CUA A-3′ | This paper | N/A |
| siSAG: 5′-CCC UCC CUU CAG AUU AUG UUA-3′ | This paper | N/A |
| siRHEB: 5′-CCC UCC CUU CAG AUU AUG UUA-3′ | This paper | N/A |
| siCUL1: 5′-GGU CGC UUC AUA AAC ACC A-3′ | This paper | N/A |
| siCUL5: 5′-UUC UCA CUU CCU ACU GAA CUG-3′ | This paper | N/A |
| qRT-PCR for Srebp1c forward: 5′-TCC AGT GGC AAA GGA GGC A-3′ | This paper | N/A |
| qRT-PCR for Srebp1c reverse: 5′-AAA GCC ACT AAG GTG CCT ACA GA-3′ | This paper | N/A |
| qRT-PCR for Sox9 forward: 5′-CGA GCA CTC TGG GCA ATC TCA-3′ | This paper | N/A |
| qRT-PCR for Sox9 reverse: 5′-GCC GTA ACT GCC AGT GTA GGT-3′ | This paper | N/A |
| qRT-PCR for Gapdh forward: 5′-AAC TTT GGC ATT GTG GAA GGG CTC-3′ | This paper | N/A |
| qRT-PCR for Gapdh reverse: 5′-TGG AAG AGT GGG AGT TGC TGT TGA-3′ | This paper | N/A |
| **Chemicals, peptides, and other reagents** | | |
| MLN4924 | APExBIO | Cat #B1036 |
| TAK243 | MedChemExpress | Cat #HY-100487 |
| 3-MA | Selleck | Cat #S2767 |
| FTI-277 | Selleck | Cat #S7465 |
| LY29400 | MedChemExpress | Cat #HY-10108 |

| Reagent/resource | Reference or source | Identifier or catalog number |
|---|---|---|
| Thymidine | Sigma-Aldrich | Cat #T1895 |
| Cycloheximide | Sigma-Aldrich | Cat #239763-M |
| Ni-NTA agarose | Qiagen | Cat #30210 |
| Anti-FLAG M2 agarose beads | Sigma-Aldrich | Cat #A2220 |
| HA peptides | APExBIO | Cat #A6010 |
| Cell Counting Kit-8 | MedChemExpress | Cat #HY-K0301 |
| **Software** | | |
| Image J | NIH | https://imagej.nih.gov/ij/ |
| GraphPad Prism 10 | GraphPad | https://www.graphpad.com/ |
| Flowjo_V10 | BD | https://www.flowjo.com/ |

## Cell culture, transfection and infection

PLC/PRF/5, SK-HEP-1, Hep3B, HeLa, HEK293 cells were obtained from ATCC. Cells were maintained in Dulbecco's modified Eagle's medium (DMEM), supplemented with 10% (v/v) fetal bovine serum (FBS) and 1% (v/v) penicillin/streptomycin. MEF cells with indicated genotypes were generated from E13.5 embryos, and then infected with Ad-GFP or Ad-Cre adenovirus to generate Ube2f WT or KO cells, respectively. MEF cells were cultured in DMEM, supplemented with 15% (v/v) FBS, 1% (v/v) non-essential amino acids and 1% (v/v) penicillin/streptomycin. Cells were transiently transfected with indicated siRNA oligos and/or plasmids using Lipofectamine 3000 reagent (Invitrogen, L3000-015), according to manufacturer's instructions.

For the lentivirus-based assays, viral particles were produced in 293T cells and subsequently infected cancer cells for 48 h in the presence of 8 μg/ml polybrene. To establish stable cell lines with knockdown or overexpression, the infected cells were subjected to selection with 1 μg/ml puromycin for 72 h.

## Plasmid construction

RHEB cDNA was subcloned into pcDNA3.1-3HA or pIRES2-EGFP vector. RHEB mutant constructs were generated with primers containing specific mutations and wild-type RHEB as a template using PCR, and then confirmed by sequencing.

## Immunoblotting and immunoprecipitation

For direct immunoblotting (IB) analysis, cells were lysed in lysis buffer (50 mM Tris-HCl, 0.15 mM NaCl, 0.1% SDS, 0.5% Sodium deoxycholate, 50 mM NaF, 1 mM $Na_3VO_4$, 1 mM EDTA, 1 mM DTT, 1% NP-40, pH 7.5), supplemented with proteasome inhibitors (APExBIO, K1007) and phosphatase inhibitors (APEx-BIO, K1012 and K1013). Protein concentrations were measured using Bradford reagent (Bio-rad, 500-0205), and samples with equal amounts were loaded onto SDS-PAGE gel, followed by IB analysis.

For immunoprecipitation (IP) analysis, cells were lysed in co-IP lysis buffer (50 mM Tris-HCl, 0.5% NP-40, 120 mM NaCl, 1 mM EDTA, pH 8.0), containing proteasome inhibitors. Cell lysates were incubated with indicated antibodies at 4 °C overnight and then incubated with protein G beads (GE Lifesciences) for additional 4 h. The immunoprecipitates were washed four times with lysis buffer, followed by IB analysis.

## In vitro binding assay

For the in vitro binding assay, 200 ng of purified His-Sumo tagged proteins along with 100 ng of GST fusion proteins were mixed in binding buffer (50 mM Tris–HCl pH 7.5, 1% Triton X-100, 150 mM NaCl, 1 mM DTT, 0.5 mM EDTA), supplemented with proteasome inhibitors. The mixture was then incubated with Ni-NTA agarose or GST beads for 4 h. Subsequently, the immuno-precipitates were washed four times with the binding buffer before subjected to IB analysis.

## EdU proliferation assay

Cells were incubated with 20 μM EdU in fresh medium for 2 h at 37 °C. After incubation, the cells were washed with PBS, trypsinized, and fixed with 4% paraformaldehyde (PFA) for 15 min at room temperature. The cells were then washed with 3% BSA for three times and treated with 0.3% Triton X-100 for 10 min, followed by washing with 3% BSA. Finally, the cells were labeled with Azide 488 for 30 min before being analyzed by FACS.

## Cell cycle synchronization

For cell cycle synchronization, cells were treated with 2 mM thymidine for 18 h, washed with PBS, and cultured in fresh medium for 9 h, followed by re-treatment with 2 mM thymidine for another 15 h before being released at specific time points. The cells were then fixed with 75% ethanol overnight and labeled with propidium iodide (PI) for FACS analysis.

## Lysosome fraction assay

Cells were washed twice with cold phosphate-buffered saline (PBS), and then centrifuged at $800 \times g$ for 2 min at 4 °C. The pellets were resuspended in hypotonic lysis buffer (10 mM HEPES, pH 7.2, 10 mM KCl, 1.5 mM $MgCl_2$, 20 mM NaF, 100 μM sodium orthovanadate, 250 mM sucrose with protease inhibitors). The cells were mechanically passed through a 23 G needle four times and then centrifuged at $500 \times g$ for 10 min at 4 °C to obtain the post-nuclear supernatants (PNSs). The PNSs were then centrifuged at $20,000 \times g$ for 2 h at 4 °C. The pellets constituted the insoluble heavy membrane fractions, while the supernatants represented the soluble light membrane fractions. The pellets were lysed in RIPA lysis buffer, followed by IB analysis.

## Cell size measurement

Cells were harvested by trypsinization and resuspended in PBS. Cell diameters were analyzed using CytoFLEX (Beckman).

## Cell growth assay

The equal numbers of cells were seeded into a 96-well plate in triplicate. Cell viability was assessed using CCK-8 reagent, according to manufacturer's instructions, and the results from three independent experiments were expressed as the fold change compared with the control.

## Gel filtration chromatography assay

Cells were lysed in CHAPS buffer (40 mM HEPES, 2 mM EDTA, 10 mM pyrophosphate, 10 mM glycerophosphate, 0.3% CHAPS, pH 7.5) containing proteasome inhibitors, and then centrifuged at 14,000 rpm for 15 min at 4 °C. The protein concentrations of lysates were adjusted to no less than 8 mg/mL. Subsequently, the lysates were passed through a 0.45-μm filter. Chromatography assays were performed in AKTA-FPLC (GE Lifesciences), and 500 μL of lysate were injected to the Superdex 200 10/300 GL column (GE Lifesciences, 17-5175-01). The proteins were then eluted with CHAPS buffer at a flow rate of 0.3 mL/min, with 500 μL per fraction, followed by IB analysis.

## In vitro kinase assay

HEK293 cells transfected with HA-S6K1 were serum-starved for 24 h, treated with 20 μM LY29400 for 1 h before being harvested, and then lysed and incubated with HA beads for 5 h at 4 °C. The immunoprecipitates were washed twice with lysis buffer, and then washed twice with wash buffer (25 mM HEPES, pH 7.4, 20 mM KCl). Finally, HA peptides were added to elute unphosphorylated HA-S6K1 protein.

For in vitro mTORC1 kinase assay, cells were harvested and lysed in CHAPS buffer and then incubated with anti-mTOR antibody to immunoprecipitate the mTOR complex. The immunoprecipitates were washed twice with CHAPS wash buffer (40 mM HEPES, 2 mM EDTA, 10 mM pyrophosphate, 10 mM glycerophosphate, 0.3% CHAPS, 150 mM NaCl, pH 7.5), and then washed twice with wash buffer. Kinase assays were performed in a kinase buffer (25 mM HEPES, 50 mM KCl, 10 mM MgCl$_2$, 250 μM ATP, pH 7.5) with mTOR immunoprecipitates and purified HA-S6K1 proteins for 20 min at 30 °C, followed by IB analysis.

## In vivo and in vitro neddylation assays

HEK293 cells were transfected with HIS-NEDD8, along with indicated plasmids for 48 h. Cells were harvested and lysed in denaturing lysis buffer (6 M guanidinium-HCl, 10 mM Tris-HCl, 0.1 M Na$_2$HPO$_4$, pH 8.0) with freshly added 10 mM β-mercaptoethanol. After ultrasonic processing, the lysates were centrifuged at 13,000 rpm for 10 min, and then incubated with Ni-NTA agarose beads for 4 h at room temperature. Beads were washed once with each of denaturing lysis buffer, Buffer A (8 M urea, 10 mM Tris-HCl, 0.1 M Na$_2$HPO$_4$, 10 mM β-mercaptoethanol, pH 8.0), Buffer B1 (8 M urea, 10 mM Tris-HCl, 0.1 M Na$_2$HPO$_4$, 10 mM β-mercaptoethanol, 0.2% Triton X-100, pH 6.3), and Buffer B2 (8 M urea, 10 mM Tris-HCl, 0.1 M Na$_2$HPO$_4$, 10 mM β-mercaptoethanol, 0.1% Triton X-100, pH 6.3) for 5 min. Neddylated proteins were eluted from beads with elution buffer (200 mM imidazole, 0.15 M Tris-HCl, 0.72 M β-mercaptoethanol, 5% SDS, 30% glycerol, pH 6.7) for 30 min at room temperature, followed by IB analysis.

The in vitro neddylation assay was performed in a 20 μL of reaction buffer, (0.2 μM ATP, 50 mM Tris-HCl (pH 7.4), 5 mM MgCl$_2$, 0.5 mM DTT, 2 μg BSA), in the presence of bacterial-purified 0.025 μM NAE, 0.3 μM NEDD8, 0.8 μM UBE2F, 200 ng SAG, and 100 ng wt-RHEB, or RHEB-K169R proteins at 37 °C for 90 min, and then followed by IB analysis.

## MD simulations

The crystal structures of RHEB structure with GTP (PDB:1XTS) were selected as the initial conformations for the simulation. The NEDD8-modified RHEB was modelled in MODELLER (version 10.1). The MD simulations were performed using AMBER20 package to optimize the predicted structures (Case et al, 2005). The general AMBER force field (GAFF), the ff14SB force field were used for the GTP, proteins, respectively. The unmodified RHEB and the NEDD8-modified RHEB systems were all immersed into a periodic box filled with the TIP3P water molecules with an extension boundary of 8 Å, and neutralized by sodium ion. The energy minimizations of each simulated system were conducted in three stages. Afterward, the system was heated to 310 K over 100 ps with the restraint of 3 kcal mol$^{-1}$ Å$^{-2}$ on the solute in the NVT ensemble. Another 100 ps MD equilibration in the NPT (P = 1 atm and T = 300 K) ensemble with a Langevin thermostat was then conducted without any restraint. At last, the MD production was conducted in the NPT ensemble with each trajectory of 100 ns duration. The SHAKE algorithm was used to restrain all hydrogen-related covalent bonds, and the time step was set to 2 fs (Zhao et al, 2022).

The relative binding energy was calculated with the MMPBSA.py script in AMBER20 using the MM/GBSA (molecular mechanics-generalized Born surface area) method. The cluster analysis was employed to extract the most representative structure from the entire simulation trajectories for the following structural analysis and 5 clusters were produced for each system. Figures are generated using Pymol (version 2.5).

## Animal work

*Pten$^{flox}$* mice (004597) and *Alb-cre* mice (003574) were purchased from Jackson laboratory. *Ube2f$^{flox}$* mice were previously generated in our laboratory (Chang et al, 2024). Animal procedures were approved by and performed in accordance with the guiding principles of the Laboratory Animal Ethics Committee of the 2$^{nd}$ Affiliated Hospital of Zhejiang University School of Medicine.

## Oil Red O staining

For Oil Red O staining, mouse livers were embedded in OCT for the preparation of frozen liver sections. 8 μm tissue sections were fixed in 10% formalin, followed by sequential washes with distilled water and 60% isopropanol. The slides were then incubated in Oil-Red-O solution for 15 min, followed by multiple washes with distilled water until no excess stain was observed. Images were captured using a section scanner (KFBIO, KF-FL-020).

## Quantitative RT-PCR

RNA was isolated from mouse livers using Trizol reagent (Invitrogen, 15596018). Subsequently, complementary DNA was reversely transcribed from RNA with the PrimeScript RT reagent

Kit (Takara, RR037A), and then subjected to real-time PCR analysis using SYBR Premix Ex Taq (TaKaRa, RR420B) on an Applied Biosystems ViiA real-time PCR system (Applied Biosystems, Waltham, MA), following the manufacturer's guidelines.

## Immunohistochemical staining

The human liver tissue microarrays consisting of 81 tumor and adjacent normal tissue were obtained from Shanghai Outdo Biotech Company (Shanghai, China). The staining intensity and percentage of positive cells were assessed for a staining score. Intensity was classified as negative (0), weak (1), moderate (2), or strong (3), while the proportion of stained cells was scored from 1 to 4 ( ≤ 10%, 1; 11–50%, 2; 51–80%, 3; and ≥81%, 4) based on percentage. The IHC score was calculated by multiplying intensity and proportion scores (Chen et al, 2020). The study has been approved by the Ethics Committee of the Shanghai Outdo Biotech Company.

Mouse livers were isolated, fixed in formalin, and embedded in paraffin. Subsequently, 5 μm tissue sections were prepared for immunohistochemical staining. After antigen retrieval, quenching, and blocking, the sections were incubated with indicated primary and secondary antibodies, followed by DAB staining. Images were captured using a section scanner (KFBIO, KF-FL-020).

## Immunofluorescence

Cultured cells were seeded onto slides and fixed with 4% paraformaldehyde (PFA) for 10 min, followed by three washes with PBS. The cells were then treated with 0.1% Triton X-100 for 5 min, followed by washing with PBS for three times. To block nonspecific binding, the slides were incubated with 3% BSA for 1 h. Subsequently, the slides were incubated with a primary antibody overnight at 4 °C, followed by incubation with fluorescent-dye-conjugated secondary antibodies and DAPI for 1 h at room temperature. Images were acquired using a confocal microscope (Nikon, Japan). Protein co-localization was quantified by calculating Pearson's correlation coefficient with plugin of ImageJ software.

## Statistical analysis

All statistical analyses were performed using GraphPad software. Kaplan–Meier survival curves were generated and compared using the log-rank test. Statistical comparisons were analyzed using the Student's *t* test or one-way analysis of variance (ANOVA) with data collected from three independent experiments. Significant differences were considered as $P < 0.05$.

# Data availability

All reagents, constructs, and cell lines will be available upon request. Source data is provided with this paper. No data amenable to large-scale data deposition were generated in this study.

The source data of this paper are collected in the following database record: biostudies:S-SCDT-10_1038-S44318-024-00353-5.

# Peer review information

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

## Acknowledgements

This work was funded in part by the National Key R&D Program of China (grants 2022YFC3401500 to YS and YZ, and 2021YFA1101000 to YS), the National Natural Science Foundation of China (grants 92253203 and U22A20317 to YS, 32471300, 92053117 and 81972591 to YZ), and Zhejiang Provincial Natural Science Foundation of China (grant LD22H300003 to YS, LZ22H160003 to YZ), and Leading Innovative and Entrepreneur Team Introduction Program of Zhejiang (grant 2022R01002 to YS), and the National Natural Science Foundation of China "Mesoscale Investigation on Tumor Substance and Energy Dynamics (grant 82188102). We appreciate the technical support provided by Chao Bi and Xiaoli Hong from the Core Facilities at Zhejiang University School of Medicine.

## Author contributions

**Fengwu Zhang**: Conceptualization; Resources; Data curation; Formal analysis; Validation; Investigation; Visualization; Methodology; Writing—original draft; Writing—review and editing. **Xiufang Xiong**: Conceptualization; Formal analysis; Validation; Investigation; Methodology; Writing—original draft; Writing—review and editing. **Zhijian Li**: Data curation; Formal analysis; Methodology. **Haibo Wang**: Data curation; Formal analysis; Methodology. **Weilin Wang**: Data curation; Formal analysis; Methodology. **Yongchao Zhao**: Conceptualization; Formal analysis; Supervision; Funding acquisition; Validation; Investigation; Visualization; Methodology; Writing—original draft; Writing—review and editing. **Yi Sun**: Conceptualization; Formal analysis; Supervision; Funding acquisition; Validation; Investigation; Methodology; Writing—original draft; Project administration; Writing—review and editing.

Source data underlying figure panels in this paper may have individual authorship assigned. Where available, figure panel/source data authorship is listed in the following database record: biostudies:S-SCDT-10_1038-S44318-024-00353-5.

## Disclosure and competing interests statement

The authors declare no competing interests.

# Expanded View Figures

**Figure EV1.  High UBE2F expression correlates with poor survival of liver cancer patients and UBE2F knockdown suppresses growth and survival by inducing autophagy.** ▶

(**A, B**) Cells stably expressing shRNAs targeting UBE2F were subjected to cell growth assay (**A**) and clonogenic survival assay (**B**). Representative images of the clonogenic assay are shown (top, **B**), and colony numbers are plotted (bottom, **B**). Data were presented as mean ± SEM from three independent experiments and analyzed by two-way ANOVA (**A**) or one-way ANOVA (**B**), respectively. The $P$ values for the comparisons were as follows: (**A**) shGFP vs. shUBE2F-1 ($P = 5E-14$) and shGFP vs. shUBE2F-2 ($P = 5.1E-14$). (**B**) shGFP vs. shUBE2F-1 ($P = 2.41E-5$) and shGFP vs. shUBE2F-2 ($P = 1.76E-4$). ***$P < 0.001$. (**C–F**) Cells transfected with the indicated siRNA were synchronized in the G1/S phase using 2 mM thymidine to block, followed by releasing with the indicated time periods. Cells were then subjected to FACS analysis (**C**) or IB with the indicated Abs (**F**), and the percentages of cells at the G0/G1 (**D**) and G2/M phases (**E**) are shown. Data are presented as mean ± SEM from three independent experiments and analyzed by two-way ANOVA. The $P$ values for the comparisons were as follows: (**D**) siCtrl vs. siUBE2F-1 ($P = 3.69E-4$ for 2 h; $6.19E-5$ for 4 h; $0.0235$ for 6 h; $0.0033$ for 10 h; $0.0012$ for 12 h) and siCtrl vs. siUBE2F-2 ($P = 0.007$ for 2 h; $6.03E-5$ for 4 h; $0.0027$ for 6 h; $0.0039$ for 10 h; $6.85E-4$ for 12 h). (**E**) siCtrl vs. siUBE2F-1 ($P = 3.92E-10$ for 4 h, $5.58E-5$ for 6 h; $1.25E-4$ for 10 h; $0.0025$ for 12 h) and siCtrl vs. siUBE2F-2 ($P = 1.47E-8$ for 4 h; $1.12E-4$ for 6 h; $1.92E-4$ for 10 h; $5.82E-4$ for 12 h). **$P < 0.01$; ***$P < 0.001$. (**G**) Cells were transfected with indicated siRNAs for 72 h, followed by IB analysis with indicated Abs. (**H**) Cells were transfected with indicated siRNAs for 72 h, autophagosomes were detected by transmission electron microscopy (TEM). The arrows indicated autophagosomes, scale bar, 1 μm. (**I**) *Ube2f$^{fl/fl}$* MEF cells were infected with Ad-GFP or Ad-Cre adenovirus for 72 h, followed by IB analysis with indicated Abs. Source data are available online for this figure.

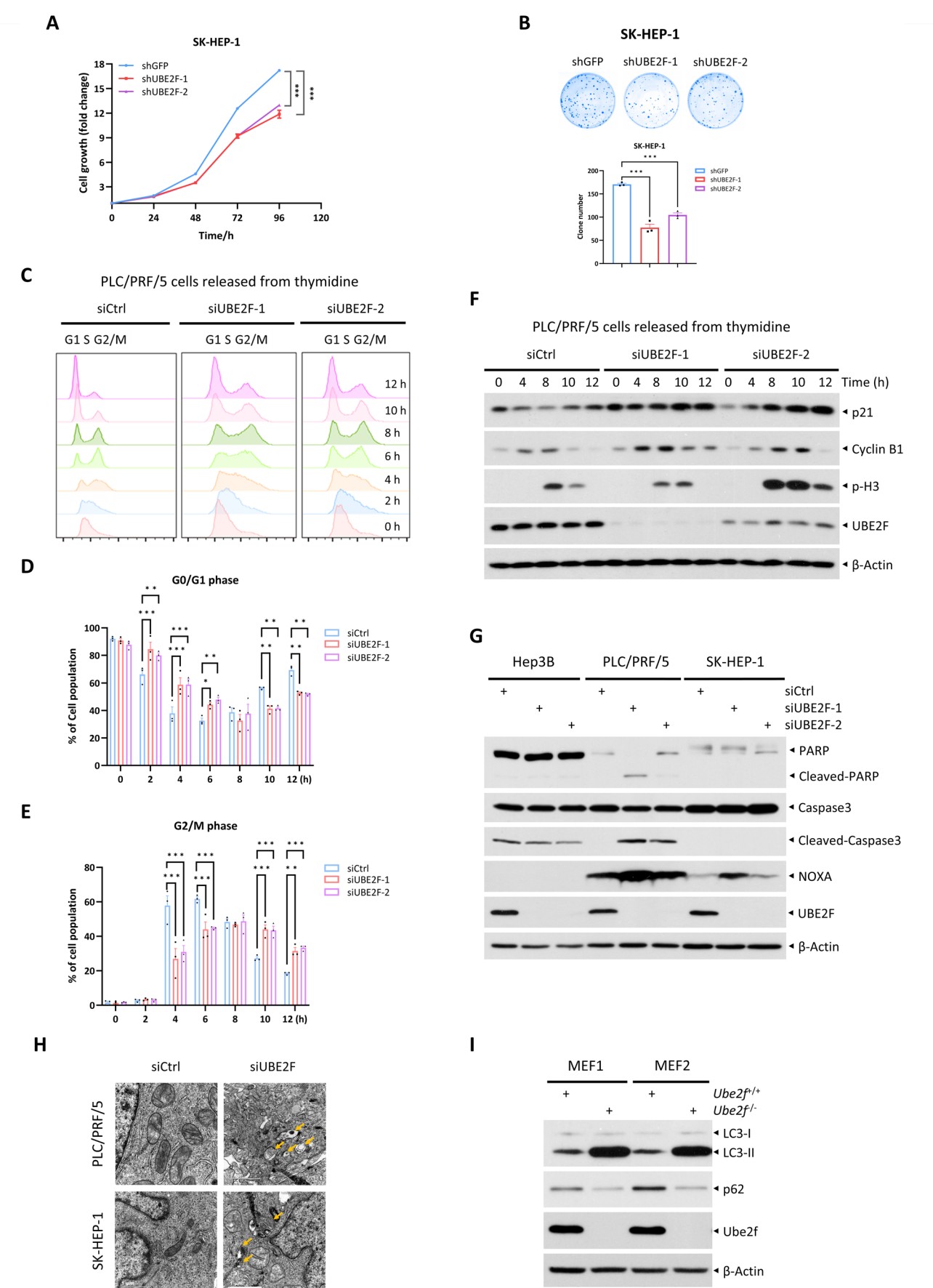

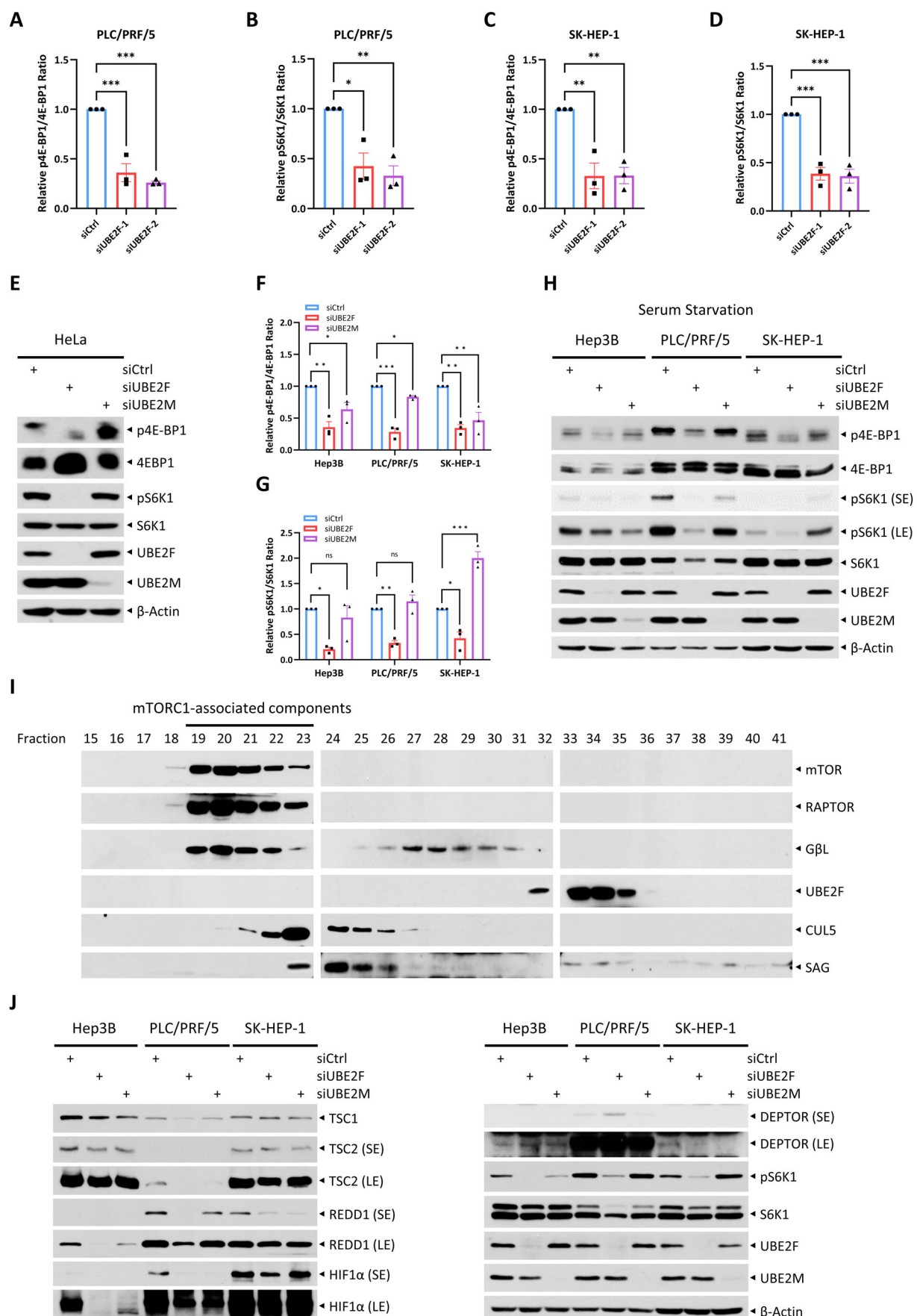

◀  **Figure EV2.  UBE2F knockdown inactivates mTORC1 activity.**

(A–D) Statistical analysis of the p4E-BP1/4E-BP1 and p-S6K1/S6K1 ratios of Fig. 2A. Data were presented as mean ± SEM from three independent experiments and analyzed by one-way ANOVA. The *P* values were as follows: (A) siCtrl vs. siUBE2F-1 (*P* = 4E-4) and siCtrl vs. siUBE2F-2 (*P* = 2E-4). (B) siCtrl vs. siUBE2F-1 (*P* = 0.0158) and siCtrl vs. siUBE2F-2 (*P* = 0.0075). (C) siCtrl vs. siUBE2F-1 (*P* = 0.0041) and siCtrl vs. siUBE2F-2 (*P* = 0.0042). (D) siCtrl vs. siUBE2F-1 (*P* = 8E-4) and siCtrl vs. siUBE2F-2 (*P* = 6E-4). \**P* < 0.05; \*\**P* < 0.01; \*\*\**P* < 0.001. (E) HeLa cells were transfected with indicated siRNAs for 72 h, followed by IB with indicated Abs. (F, G) Statistical analysis of the p4E-BP1/4E-BP1 and p-S6K1/S6K1 ratios of Fig. 2B. Data were presented as mean ± SEM from three independent experiments and analyzed by one-way ANOVA. The *P* values were as follows: (F) Hep3B siCtrl vs. siUBE2F (*P* = 0.0022) and siCtrl vs. siUBE2M (*P* = 0.0317); PLC/PRF/5 siCtrl vs. siUBE2F (*P* = 1.14E-5) and siCtrl vs. siUBE2M (*P* = 0.025); SK-HEP-1 siCtrl vs. siUBE2F (*P* = 0.0019) and siCtrl vs. siUBE2M (*P* = 0.0054). (G) Hep3B siCtrl vs. siUBE2F (*P* = 0.0108) and siCtrl vs. siUBE2M (*P* = 0.6087); PLC/PRF/5 siCtrl vs. siUBE2F (*P* = 0.0016) and siCtrl vs. siUBE2M (*P* = 0.3603); SK-HEP-1 siCtrl vs. siUBE2F (*P* = 0.0127) and siCtrl vs. siUBE2M (*P* = 8E-4). \**P* < 0.05; \*\**P* < 0.01; \*\*\**P* < 0.001; ns: no significant. (H) Cells were transfected with indicated siRNAs for 48 h, and then serum starved for 24 h before being harvested for IB analysis with indicated Abs. (I) SK-HEP-1 cells were lysed in CHAPS buffer. After being adjusted to 10 mg/mL of protein concentration, the lysate was passed through a 0.45 μm filter. 500 μL of the lysate was injected into the Superdex 200 10/300 GL column, and equal amounts were then collected for IB analysis. (J) Cells were transfected with indicated siRNAs for 72 h and then subjected to IB analysis with indicated Abs. SE short exposure, LE long exposure. Source data are available online for this figure.

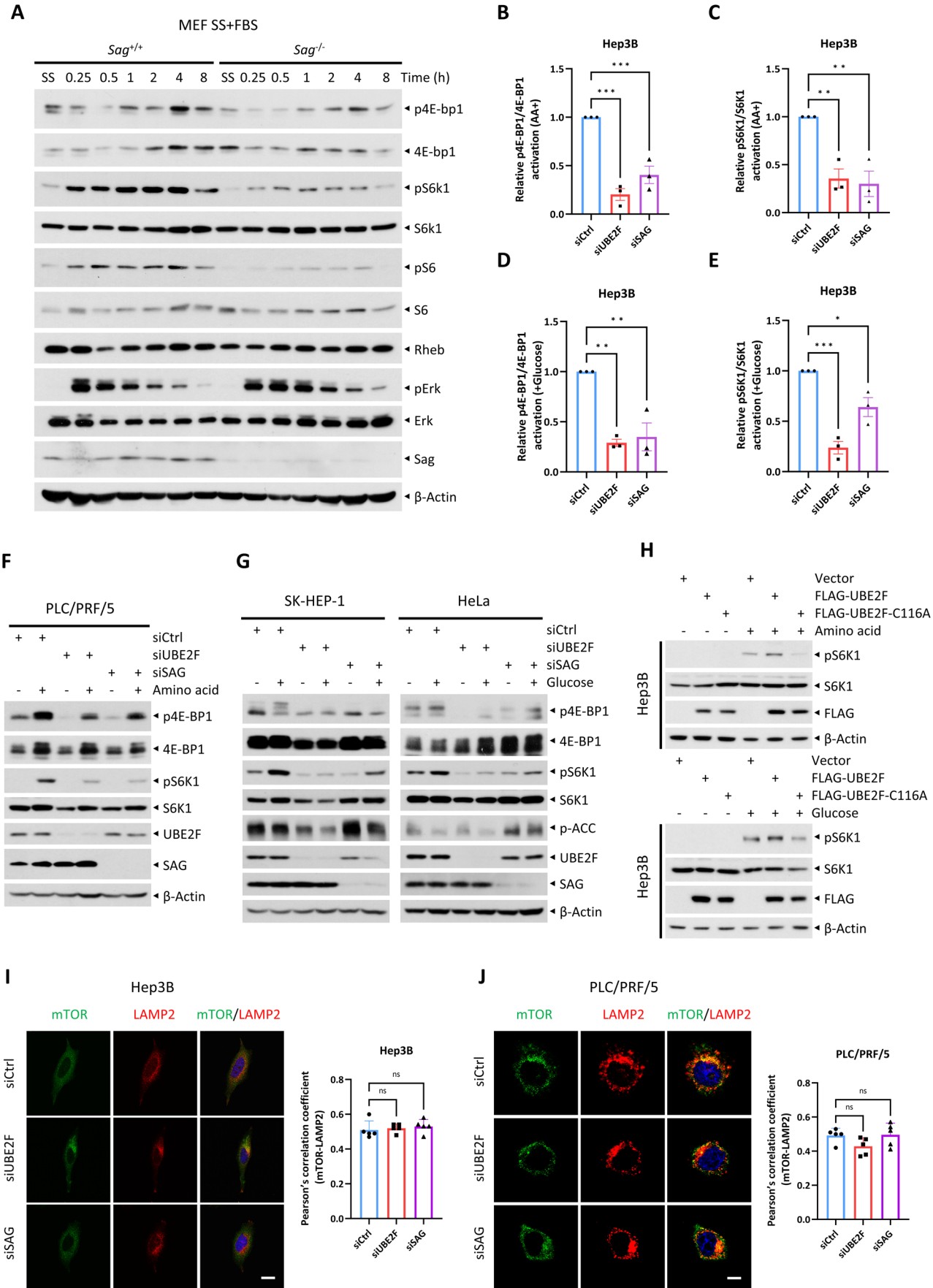

◄  **Figure EV3.  The UBE2F-SAG axis regulates mTORC1 activity.**

(A) $Sag^{+/+}$ and $Sag^{fl/fl}$ MEF cells were infected with Ad-Cre adenovirus for 72 h, and then serum starved for 24 h, followed by serum re-suppy for indicated time periods before being harvested for IB analysis with indicated Abs. (B–E) Statistical analysis of the p4E-BP1/4E-BP1 and p-S6K1/S6K1 ratios from Fig. 3B,C. Data were presented as mean ± SEM from three independent experiments and analyzed by one-way ANOVA. The *P* values were as follows: (B) siCtrl vs. siUBE2F ($P = 2E\text{-}4$) and siCtrl vs. siSAG ($P = 9E\text{-}4$). (C) siCtrl vs. siUBE2F ($P = 0.0063$) and siCtrl vs. siSAG ($P = 0.0042$). (D) siCtrl vs. siUBE2F ($P = 0.0016$) and siCtrl vs. siSAG ($P = 0.0026$). (E) siCtrl vs. siUBE2F ($P = 3E\text{-}4$) and siCtrl vs. siSAG ($P = 0.0138$). *$P < 0.05$; **$P < 0.01$; ***$P < 0.001$. (F) PLC/PRF/5 cells were transfected with indicated siRNAs for 72 h, and then deprived of amino acid for 50 min, followed by re-stimulation with amino acid for 15 min before being harvested for IB analysis with indicated Abs. (G) Cells were transfected with indicated siRNAs for 72 h, and then starved of glucose for 6 h, followed by restimulation with glucose for 20 min before being harvested for IB analysis with indicated Abs. (H) Hep3B cells transfected with 3 μg of WT-RHEB or the enzymatic-dead mutant UBE2F-C116A were subjected to amino acid or glucose deprivation, followed by refeeding before being harvested for IB analysis with the indicated Abs. (I, J) Hep3B and PLC/PRF/5 cells were transfected with indicated siRNAs for 48 h, and then immunostained for mTOR (green) and LAMP2 (red). The statistical analysis of the co-localization of mTOR and LAMP2 was performed by calculating Pearson's correlation coefficient. Data were presented as mean ± SD from five random fields and analyzed by one-way ANOVA. ns: no significant. Scale bars, 10 μm. Source data are available online for this figure.

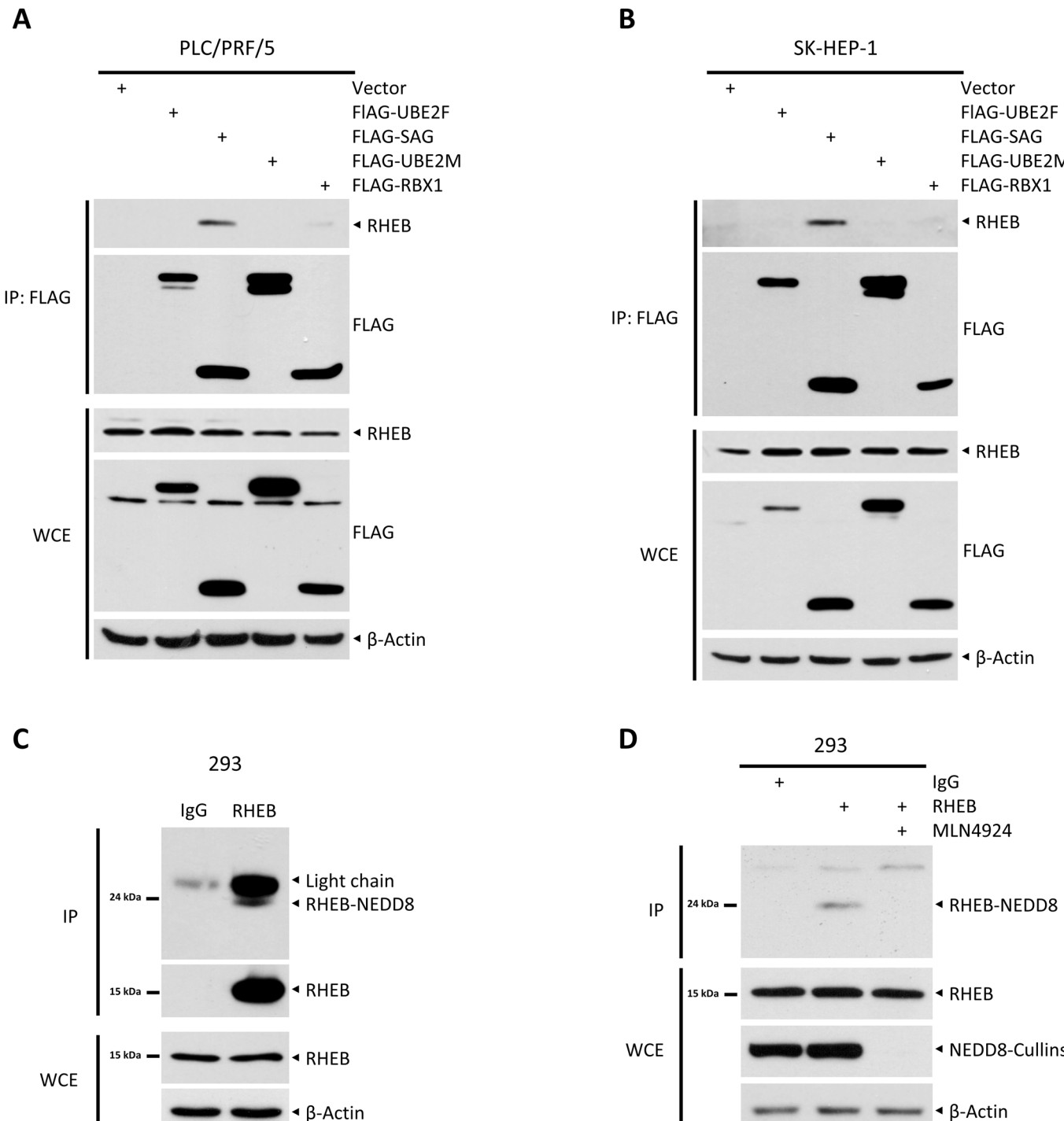

**Figure EV4. The UBE2F-SAG axis promotes RHEB neddylation.**

(A, B) PLC/PRF/5 (A) and SK-HEP-1 (B) cells were transfected with indicated plasmids for 48 h, and then subjected to IP with FLAG beads, followed by IB analysis with indicated Abs. (C) HEK293 cell lysates were incubated with anti-RHEB Ab or normal IgG as a control, followed by IB analysis with anti-NEDD8 Ab to detect endogenous RHEB neddylation. (D) Cells were left untreated or treated with 1 μM MLN4924 for 24 h and then subjected to IP with anti-RHEB Ab, along with normal IgG, followed by IB analysis with anti-NEDD8 Ab to detect endogenous RHEB neddylation. WCE whole cell extract. Source data are available online for this figure.

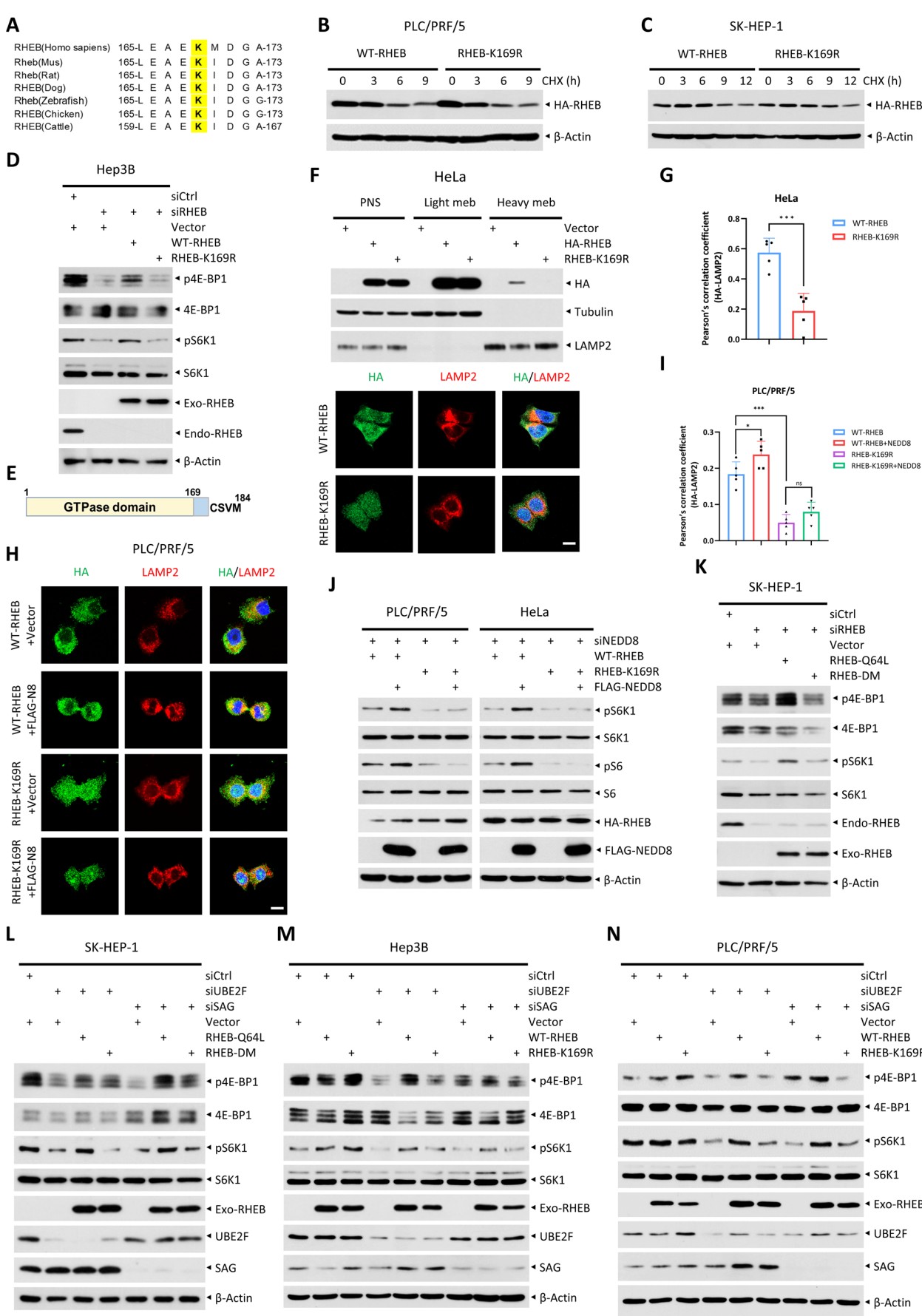

**Figure EV5.  Neddylation K169R mutant loses RHEB activity with altered lysosome localization.**

(A) RHEB K169 residues were evolutionarily conserved and highlighted. (B, C) Cells were transfected with 3 µg HA-tagged wild type RHEB (WT-RHEB) or RHEB-K169R mutant for 48 h, and then treated with 100 µg/mL CHX for indicated time periods, followed by IB analysis with indicated Abs. (D) Hep3B cells were simultaneously transfected with siRNA targeting RHEB 3′UTR region along with 3 µg exogenous RHEB plasmids for 72 h, followed by IB analysis with indicated Abs. (E) RHEB domain structure. (F, G) HeLa cells were transfected with 15 µg HA-WT-RHEB or HA-RHEB-K169R mutant per 15 cm dish for 48 h, and then subjected to IB analysis with indicated Abs after isolation of heavy membrane and light membrane fractions (top, **F**), or immunofluorescent labeling of HA (green) and LAMP2 (red) (bottom, **F**). The statistical analysis of the co-localization of HA and LAMP2 was performed by calculating Pearson's correlation coefficient (**G**). Data were presented as mean ± SD from five random fields and analyzed by Student's *t* test, $P = 4E-4$. ***$P < 0.001$. PNS: post-nuclear supernatants. Scale bar, 10 µm. (H–J) PLC/PRF/5 and HeLa cells were co-transfected with 1.5 µg of RHEB and 2 µg of NEDD8 plasmids for 48 h. Cells were subjected to immunofluorescent labeling of HA (green) and LAMP2 (red) (**H**) or IB analysis (**J**). The statistical analysis of the co-localization of HA and LAMP2 was performed by calculating Pearson's correlation coefficient (**I**). Data were presented as mean ± SD from five random fields and analyzed by Student's *t* test. The *P* values for the comparisons were as follows: WT-RHEB vs. WT-RHEB + NEDD8 ($P = 0.0406$); WT-RHEB vs. RHEB-K169R ($P = 7.46E-5$); RHEB-K169R vs. RHEB-K169R + NEDD8 ($P = 0.0888$). Scale bar, 10 µm, *$P < 0.05$; ***$P < 0.001$; ns: no significant. (K) SK-HEP-1 cells were simultaneously transfected with siRNA targeting the 3′UTR region of RHEB, along with 3 µg of exogenous RHEB plasmids for 72 h, followed by IB analysis with the indicated Abs. (L–N) Cells were transfected with indicated siRNA and plamids for 72 h, followed by IB analysis with indicated Abs. RHEB-DM: RHEB-Q64L-K169R (**K, L**). Source data are available online for this figure.

                                            

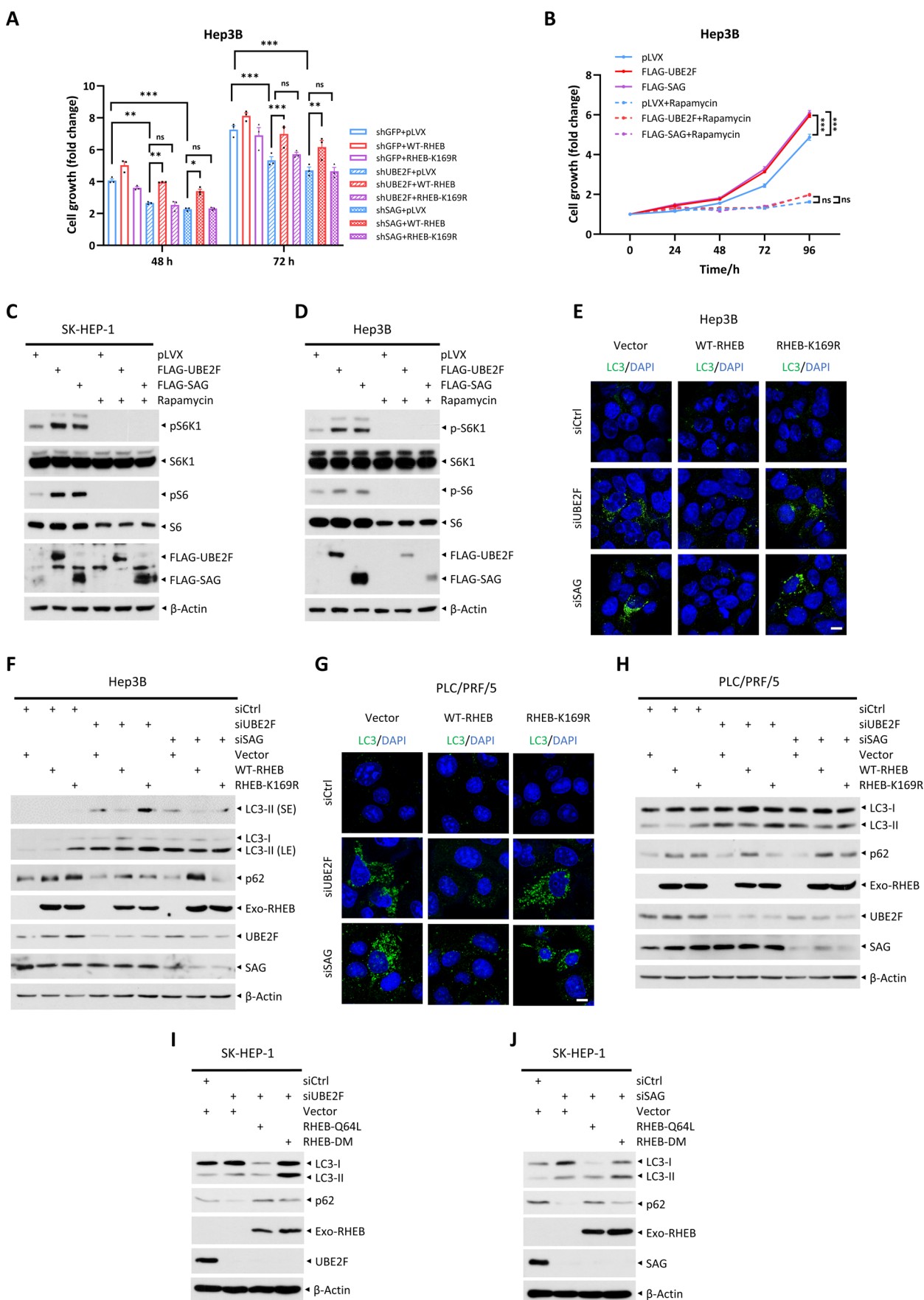

◀  **Figure EV6.   RHEB-K169R mutant fails to rescue altered growth phenotypes induced by knockdown of UBE2F or SAG.**

(**A**) Hep3B cells stably expressing shRNA and plasmids were subjected to cell growth assay. Data were presented as mean ± SEM from three independent experiments and analyzed by two-way ANOVA. The $P$ values for the various comparisons were as follows: shGFP+pLVX vs. shUBE2F+pLVX ($P = 0.0036$ for 48 h and 3.57E-5 for 72 h); shGFP+pLVX vs. shSAG+pLVX ($P = 1.11$E-4 for 48 h and 1.16E-7 for 72 h); shUBE2F+pLVX vs. shUBE2F+WT-RHEB ($P = 0.0084$ for 48 h and 4.48E-4 for 72 h); shUBE2F +pLVX vs. shUBE2F+RHEB-K169R ($P = 0.9999$ for 48 h and 0.9618 for 72 h); shSAG+pLVX vs. shSAG+WT-RHEB ($P = 0.035$ for 48 h and 0.0024 for 72 h); shSAG +pLVX vs. shSAG+RHEB-K169R ($P = 0.9999$ for both 48 h and 72 h). *$P < 0.05$; **$P < 0.01$; ***$P < 0.001$; ns: no significant. (**B**) Hep3B cells were infected with lentivirus-based FLAG-UBE2F or FLAG-SAG plasmids, cultured with 2.5% FBS, and then treated with or without 1 μM rapamycin, followed by cell growth assay. Data were presented as mean ± SEM from three independent experiments and analyzed by two-way ANOVA. The $P$ values for the comparisons were as follows: pLVX vs. FLAG-UBE2F ($p = 1.9924$E-11); pLVX vs. FLAG-SAG ($P = 1.9904$E-11); pLVX+Rapamycin vs. FLAG-UBE2F+Rapamycin ($P = 0.0867$); pLVX+Rapamycin vs. FLAG-SAG+Rapamycin ($P = 0.0858$). ***$P < 0.001$; ns: no significant. (**C**, **D**) Hep3B and SK-HEP-1 cells were infected with the indicated expressing lentivirus and then treated with 100 nM rapamycin for 24 h in 2.5% FBS before being harvesed for IB analysis. (**E–H**) Hep3B and PLC/PRF/5 cells were transfected with the indicated siRNA and plasmids for 48 h, followed by co-staining of LC3 (green) and DAPI (blue) (**E**, **G**) or IB analysis with the indicated Abs (**F**, **H**). Scale bar, 10 μm. (**I**, **J**) Cells were transfected with indicated siRNA and plasmids for 72 h, followed by IB analysis with indicated Abs. Source data are available online for this figure.

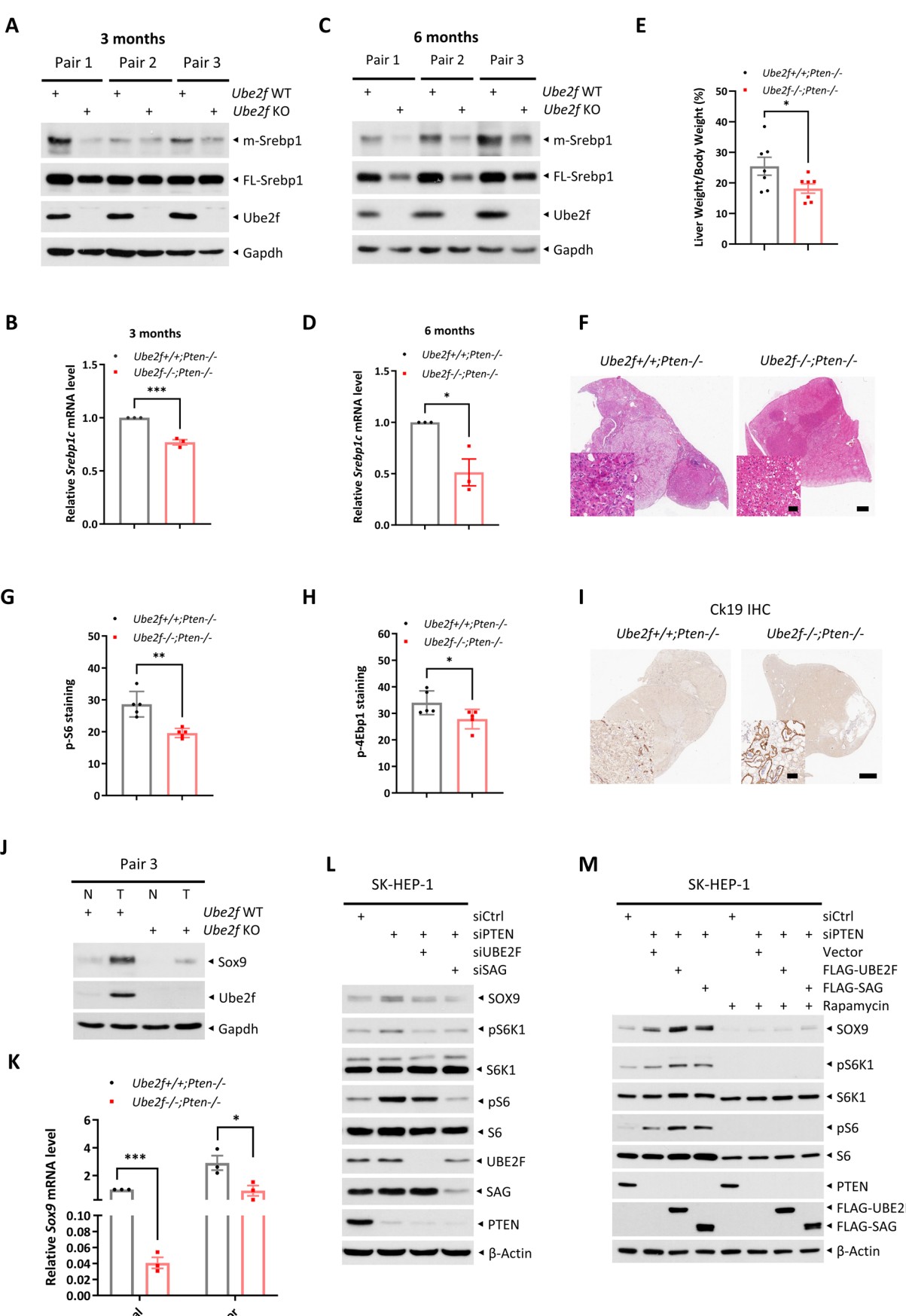

◀ **Figure EV7.** *Ube2f* **knockout attenuates liver steatosis and tumorigenesis in vivo.**

(A–D) Liver tissues were isolated from paired *Ube2f + / +;Pten-/-* (Ube2f WT) and *Ube2f-/-;Pten-/-* (Ube2f KO) mice at the ages of 3 and 6 months and then subjected to IB analysis with indicated Abs (A&C) or qRT-PCR analysis (B&D). Data were shown as mean ± SEM and analyzed by Student's t test. n = 3 for each genotype. The P values were 7E-4 for (B) and 0.0199 for (D). *P < 0.05; ***P < 0.001. (E) The ratios of liver/body weight of Ube2f WT and KO mice at 12-month old were shown as mean ± SEM and analyzed by Student's t test. P = 0.0471, n = 7 for each genotype; *P < 0.05. (F) H&E staining of liver tissues from 12-month-old Ube2f WT and KO mice. Scale bars, 1 mm and 40 μm (inset), respectively. (G, H) Quantification of p-S6 (G) and p-4Ebp1 (H) staining from Fig. 7E. Data were shown as mean ± SD from five random fields and analyzed by Student's t test. The P values were 0.0014 for (G) and 0.0449 for (H). *P < 0.05; **P < 0.01. (I) Ck-19 staining of liver sections from 12-month-old Ube2f WT and KO mice. Scale bars, 2 mm and 80 μm (inset), respectively. (J, K) Liver tissues from 12-month-old Ube2f WT and KO mice were harvested for IB (J) or qRT-PCR analysis (K). Data were shown as mean ± SEM and analyzed by Student's t test. n = 3 for each genotype. The P values were as follows: normal tissue (Ube2f WT vs. KO, P = 1.7E-8) and tumor tissue (Ube2f WT vs. KO, P = 0.0352). *P < 0.05; ***P < 0.001. (L, M) SK-HEP-1 cells were transfected with the indicated siRNAs (L) or with 3 μg of the indicated plasmids (M) for 72 h. The cells were then treated with or without 100 nM rapamycin for 48 h before being harvested for IB with indicated Abs. Source data are available online for this figure.

