## [Peer Review File · The EMBO Journal]

RHEB neddylation by the UBE2F-SAG axis enhances mTORC1 activity and aggravates liver tumorigenesis

Fengwu Zhang, Xiufang Xiong, Zhijian Li, Haibo Wang, Weilin Wang, Yongchao Zhao, and Yi Sun

Corresponding authors: Yi Sun (yisun@zju.edu.cn) , Yongchao Zhao (yongchao@zju.edu.cn)

Review Timeline:

Submission Date:	2nd Jun 24
Editorial Decision:	6th Jul 24
Revision Received:	31st Oct 24
Editorial Decision:	18th Nov 24
Revision Received:	27th Nov 24
Accepted:	4th Dec 24

Editor: Daniel Klimmeck

Transaction Report:

Dear Dr Sun,

Thank you for submitting your manuscript for consideration by the EMBO Journal, as well as for your patience with our feedback at this time of the year. Your work has now been seen by three referees with expertise in epithelial cancer signaling and PTMs, whose comments are shown below.

Given the overall interest stated and broader angle of your findings, we are able to invite you to revise your manuscript experimentally to address the referees' comments. I need to stress though that we do require strong support from the referees on a revised version of the study in order to move on to publication of the work.

I would appreciate if you could contact me during the next weeks for exchange e.g. a video call to discuss your perspective on the comments and potential plan for revisions.

Please feel free to contact me if you have any questions or need further input on the referee comments.

When submitting your revised manuscript, please carefully review the instructions below.

Please feel free to approach me any time should you have additional questions related to this.

Thank you for the opportunity to consider your work for publication.

I look forward to your revision.

Best regards,

Daniel Klimmeck

Daniel Klimmeck, PhD
Senior Editor
The EMBO Journal

Instruction for the preparation of your revised manuscript:

- 1) a .docx formatted version of the manuscript text (including legends for main figures, EV figures and tables). Please make sure that the changes are highlighted to be clearly visible.
- 2) individual production quality figure files as .eps, .tif, .jpg (one file per figure).
- 3) a .docx formatted letter INCLUDING the reviewers' reports and your detailed point-by-point response to their comments. As part of the EMBO Press transparent editorial process, the point-by-point response is part of the Review Process File (RPF), which will be published alongside your paper.
- 4) a complete author checklist, which you can download from our author guidelines ([https://wol-prod-cdn.literatumonline.com/pb-assets/embo-site/Author Checklist%20-%20EMBO%20J-1561436015657.xlsx](https://wol-prod-cdn.literatumonline.com/pb-assets/embo-site/Author%20Checklist%20-%20EMBO%20J-1561436015657.xlsx)). Please insert information in the checklist that is also reflected in the manuscript. The completed author checklist will also be part of the RPF.
- 5) Please note that all corresponding authors are required to supply an ORCID ID for their name upon submission of a revised manuscript.
- 6) It is mandatory to include a 'Data Availability' section after the Materials and Methods. Before submitting your revision, primary datasets produced in this study need to be deposited in an appropriate public database, and the accession numbers and database listed under 'Data Availability'. Please remember to provide a reviewer password if the datasets are not yet public (see

<https://www.embopress.org/page/journal/14602075/authorguide#datadeposition>).

7) Our journal encourages inclusion of *data citations in the reference list* to directly cite datasets that were re-used and obtained from public databases. Data citations in the article text are distinct from normal bibliographical citations and should directly link to the database records from which the data can be accessed. In the main text, data citations are formatted as follows: "Data ref: Smith et al, 2001" or "Data ref: NCBI Sequence Read Archive PRJNA342805, 2017". In the Reference list, data citations must be labeled with "[DATASET]". A data reference must provide the database name, accession number/identifiers and a resolvable link to the landing page from which the data can be accessed at the end of the reference. Further instructions are available at .

8) At EMBO Press we ask authors to provide source data for the main and EV figures. Our source data coordinator will contact you to discuss which figure panels we would need source data for and will also provide you with helpful tips on how to upload and organize the files.

Numerical data can be provided as individual .xls or .csv files (including a tab describing the data). For 'blots' or microscopy, uncropped images should be submitted (using a zip archive or a single pdf per main figure if multiple images need to be supplied for one panel). Additional information on source data and instruction on how to label the files are available at .

9) We replaced Supplementary Information with Expanded View (EV) Figures and Tables that are collapsible/expandable online (see examples in <https://www.embopress.org/doi/10.15252/emboj.201695874>). A maximum of 5 EV Figures can be typeset. EV Figures should be cited as 'Figure EV1, Figure EV2' etc. in the text and their respective legends should be included in the main text after the legends of regular figures.

11) For data quantification: please specify the name of the statistical test used to generate error bars and P values, the number (n) of independent experiments (specify technical or biological replicates) underlying each data point and the test used to calculate p-values in each figure legend. The figure legends should contain a basic description of n, P and the test applied. Graphs must include a description of the bars and the error bars (s.d., s.e.m.).

We realize that it is difficult to revise to a specific deadline. In the interest of protecting the conceptual advance provided by the work, we recommend a revision within 3 months (4th Oct 2024). Please discuss the revision progress ahead of this time with the editor if you require more time to complete the revisions.

Referee #1:

In this manuscript, Zhang et al. propose the UBE2F-SAG neddylation ligase complex as a new regulator of mTORC1 activity. In particular, the UBE2F-SAG would activate mTORC1 thus leading to autophagy inhibition and tumorigenesis in liver cancer mouse models. Mechanistically, the authors propose that UBE2F-SAG neddylates RHEB, a well-known mTORC1 activator, thus promoting its lysosome localization and GTP-binding. Localization of RHEB at the lysosomes would then sustain mTORC1 activity.

Altogether this is an interesting work as it may help identify a novel signaling mechanism regulated by neddylation, and uncover new pathways driving liver tumorigenesis.

However, some of the experiments proposed in the manuscript display some weaknesses which need to be addressed by the authors. Moreover, the quality of some experiments needs to be improved.

In particular, in virtually all figures the signal of the western blot analyses is saturated, thus making quantitative analyses extremely difficult. Non-saturated western blots should be therefore used in the figures.

In addition, in virtually all figures western blot panels have been cropped very close to the bands, thus making problematic the assessment of potential post-translational modifications (PTMs) of the analyzed proteins.

Figure legends also need to be improved, very often they lack experimental detail that is necessary for the correct interpretation of the results.

Herein are some of the concerns that should be addressed by the authors.

Major concerns:

Figure 1L. Autophagy inhibitors (e.g., Bafilomycin) should be used as control to rescue autophagy induction upon UBE2F depletion.

Figure EV1G. The authors propose that UBE2F depletion does not impact apoptosis. However, the results obtained in the 3 cell lines examined seem to indicate the opposite. Indeed, Hep3B cells display high levels of apoptotic markers both in siCTRL and siUBE2F cells. In addition (as also mentioned by the authors), PLC/PRF/5 cells display apoptosis activation in the siUBE2F cells.

Figure 2B. The authors propose that UBE2F depletion would reduce mTORC1 activity thus reducing phosphorylation of its substrates (e.g., pS6K1 and p4E-BP1). However, in this figure it appears that siUBE2F triggers downregulation of S6K1 total levels in PLC/PRF/5 cells, whereas in SK-HEP-1 cells siUBE2F has no effect on the levels of pS6K1 and p4E-BP1. The authors should clarify this discrepancy.

Figure EV2B. Similar issue is present in this figure. pS6K1 levels do not change in siUBE2F Hep3B cells. In addition, pS6K1 levels seem to be affected upon UBE2M depletion thus contradicting the hypothesis of the authors.

In particular, pS6K1 seems to be downregulated in siUBE2M Hep3B cells, whereas it seems to be upregulated in siUBE2M Sk-Hep-1 cells.

Figure 2E. Western blot for UBE2F and SAG should be included to assess whether they can interact (either directly or indirectly) with the mTORC active complex.

Figure EV2C. Co-immunoprecipitation experiments should be performed to confirm that mTORC1 does not associate with UBE2F-SAG.

Figure EV2D. The results shown here seem to contradict the authors' claim:

"In fact, we found that UBE2F knockdown did not significantly affect the levels of few upstream regulators of mTORC1, including HIF1 α , DEPTOR, TSC1, or TSC2, although REDD1 levels were reduced in a cell line-dependent manner (Fig. EV2D)"
Indeed, besides REDD1, also HIF1, DEPTOR, and TSC1-2 appear to be downregulated in the siUBE2F cells. This result may suggest that UBE2F depletion may alter mTORC1 activity by affecting the levels of its upstream regulators. The authors should address this point.

Figures 3B-D. The authors should confirm these results using UBE2F-SAG inactive mutant.

Figures 4A-B. Western blot for UBE2F and mTOR should be included to check whether these proteins could form a complex.

Figure 4C. GST-only should be included as negative control.

Figures 4G-H. Neddylation assays should be analyzed probing the neddylated substrate with a NEDD8 antibody.

Figure 6. Importantly, the results of the functional analyses shown in this figure should also include genetic or pharmacological inactivation of mTOR. This is extremely important to demonstrate that, indeed, manipulation of the UBE2F-SAG-RHEB pathway affects the biology of the cells via mTOR regulation.

Figure 7I. Similarly, in this figure genetic or pharmacological inactivation of mTOR should be performed.

Figure 7 A-G. Quantification and statistical analyses should be included for all these experiments. Without quantitative analyses it is difficult to assess the biological relevance of this pathway.

Minor concerns:

Figures 4D and 4F. It is not clear what type of NEDD8 antibodies (NEDD8-cullins and NEDD8-exo-Rheb) have been used in this experiment. Authors should clarify.

Figure 4H. RHEB-K169R mutant is used in this experiment without a previous explanation. Author should either reorganize the order of the figures or comment in the main text how they identified this mutant before figure 5. In the current version of the manuscript this mutant is described only afterwards, i.e., in figure 5.

Referee #2:

Summary:

In this manuscript, Zhang et al identified RHEB as a novel substrate for neddylation by the UBE2F-SAG E2-E3 axis, which modifies RHEB at the K169 site to enhance its lysosome localization and GTP-binding affinity, thereby activating mTORC1. They also demonstrated the role of UBE2F-SAG in regulating mTORC1 activity and its potential in treating liver diseases. Overall, this manuscript provides detailed mechanistic insights into how neddylation influences mTORC1 activity. This work is innovative and significant. Data set support the proposed hypothesis. Several major and minor points need to be addressed:

Major points:

1. In Figure 6, the author suggests that overexpression of RHEB could counteract mTORC1 inactivation, growth suppression, cell size reduction, and autophagy induction in UBE2F-knockdown cells. This implies that WT RHEB might still activate mTORC1 in the absence of proper neddylation. Further experiments are needed to clarify whether overexpressed WT RHEB can compensate for the loss of UBE2F and whether this activation occurs independently of neddylation.
2. What are the expression levels of RHEB under conditions of serum starvation and refeeding? Given that some studies suggest energy status may directly affect RHEB levels (<https://doi.org/10.1038/s41419-024-06808-1>), it is important to determine if there are any changes in RHEB expression.
3. To thoroughly evaluate the role of UBE2F in tumor cell proliferation, establishing stable cell lines with UBE2F knockdown is recommended, particularly for clonogenic survival assays.
4. Similarly, the hypothesis testing in Figure 6 would benefit from the use of stable cell lines expressing the RHEB-K169R mutant. This approach would allow for a more consistent and controlled study of the mutant's effects.
5. The author notes that RHEB can undergo multiple post-translational modifications. Is there a potential interplay between neddylation and other modifications, such as ubiquitylation?
6. The author mentions that targeting the UBE2F-SAG axis may have therapeutic advantages for certain liver diseases. Has the author tested whether the neddylation inhibitor MLN4924 could enhance the effectiveness of current anti-cancer therapies in tumors with high UBE2F expression?

Minor Points:

1. Please ensure that the manuscript format is consistent throughout the document.
2. Ensure that the height of the western blot bands is consistent across all figures. For example, compare the band height in Figure 4A IP RHEB with IP SAG.
3. Please provide β -actin controls in all panels. Several panels are missing these controls, such as in Figure 4A compared to 4B.

Referee #3:

In this paper, Zhang et al found that UBE2F, a Nedd8 (N8) E2 that works together with the E3 SAG/RBX2, Neddylates RHEB, leading to mTORC1 activation, increased cell size and proliferation (and inhibiting autophagy), promoting liver tumor growth. These authors thus identified RHEB as a novel substrate for UBE2F/SAG (different than the well-known Cul5 substrate). The Neddylation of RHEB leads to its lysosomal membrane localization and elevated GTP binding (ie active RHEB), important for activation of mTORC1 at the lysosomal membrane.

Overall, this is an interesting, extensive, systematic and carefully executed study that spans in vitro, in cellulo and in vivo studies (the latter using ube2f conditional knockout mice in the liver), as well as analysis of human liver tissues and tumors.

Critique:

1. Much of the work presented relies on Western blots (WB) and IPs, which is logical, in order to study the biochemical processes involved. The major problem, however, is that none of these biochemical analyses (WB, IP) are quantified, with no statistics provided. Moreover, in most cases normalization of the data is required, but not provided. Eg pS6 needs to be normalized to total S6, and p4E-BP normalized to total 4E-BP. For example, in Fig 2C and Fig 3B,C: KO or KD or UBE2F or SAG resulted in reduction of p4E-BP, but so does total 4E-BP. So, is mTORC1 really inhibited by lack of UBE2F?
 2. Likewise, immunofluorescence (IF) studies of RHEB localization to lysosomes following Neddylation show only 1-2 cells, with no statistics provided (eg Pearson coefficient to prove colocalization with lysosomes)
- So, overall, while these results appear compelling (based on one blot per experiment or 1-2 cells shown), lack of quantitation and statistics is problematic and makes it difficult to have confidence in some of the conclusions drawn.
3. How does Neddylation of RHEB target it to the lysosomal membrane? Is Neddylation of Cul5 also targets it to the lysosomal membrane?
 4. And how does Neddylation of RHEB lead to increase of its GTP binding (active form)?
 5. Fig EV4D (Neddylation of RHEB in 293 cells): the faint band shown to represent Neddylation is not convincing, especially as it is not reduced in the presence of the Neddylation inhibitor MLN4924.
 6. How does UBE2F promote cell proliferation without affecting the cell cycle?
 7. Analysis of liver tumor tissues: were they graded blindly?

Responses to the comments from three insightful reviewers

Referee #1:

In this manuscript, Zhang et al. propose the UBE2F-SAG neddylation ligase complex as a new regulator of mTORC1 activity. In particular, the UBE2F-SAG would activate mTORC1 thus leading to autophagy inhibition and tumorigenesis in liver cancer mouse models. Mechanistically, the authors propose that UBE2F-SAG neddylates RHEB, a well-known mTORC1 activator, thus promoting its lysosome localization and GTP-binding. Localization of RHEB at the lysosomes would then sustain mTORC1 activity.

Altogether this is an interesting work as it may help identify a novel signaling mechanism regulated by neddylation, and uncover new pathways driving liver tumorigenesis.

Thanks for your positive comments and brief summary of our work.

However, some of the experiments proposed in the manuscript display some weaknesses which need to be addressed by the authors. Moreover, the quality of some experiments needs to be improved.

We have addressed your concerns point-by-point below.

In particular, in virtually all figures the signal of the western blot analyses is saturated, thus making quantitative analyses extremely difficult. Non-saturated western blots should be therefore used in the figures.

In addition, in virtually all figures western blot panels have been cropped very close to the bands, thus making problematic the assessment of potential post-translational modifications (PTMs) of the analyzed proteins.

Figure legends also need to be improved, very often they lack experimental detail that is necessary for the correct interpretation of the results.

We appreciated your detailed comments and have made suggested changes throughout all figure panels accordingly, along with experimental details in our figure legends.

Herein are some of the concerns that should be addressed by the authors.

Major concerns:

Figure 1L. Autophagy inhibitors (e.g., Bafilomycin) should be used as control to rescue autophagy induction upon UBE2F depletion.

Given the request by the reviewer was to rescue autophagy induction upon UBE2F depletion, we have used 3-MA, an autophagy induction inhibitor (instead of Bafilomycine) for this experiment. Our newly generated data showed that 3-MA treatment indeed caused significant alleviation of LC3-II accumulation induced by UBE2F knockdown, supporting our conclusion (see below and Fig. 1L).

Figure EV1G. The authors propose that UBE2F depletion does not impact apoptosis. However, the results obtained in the 3 cell lines examined seem to indicate the opposite. Indeed, Hep3B cells display high levels of apoptotic markers both in siCTRL and siUBE2F cells. In addition (as also mentioned by the authors), PLC/PRF/5 cells display apoptosis activation in the siUBE2F cells.

We agreed with the reviewer that there are some cell-line dependent variations in apoptosis induction, which in some cases, was due to transfection reagent (e.g. in Hep3B cells). We acknowledged that UBE2F knockdown induced modest apoptosis in PLC/PRF/5 cells, but not at all in SK-HEP-1 and Hep3B cells. We, therefore, concluded that the major cause of growth inhibition upon UBE2F knockdown was not due to apoptosis in liver cancer cells, unlike robust apoptosis induction seen in lung cancer cells, as we previously reported (CCR, 2017, PMID: 27591266). We have now changed the language in the text to reflect this point (Pg 7).

Figure 2B. The authors propose that UBE2F depletion would reduce mTORC1 activity thus reducing phosphorylation of its substrates (e.g., pS6K1 and p4E-BP1). However, in this figure it appears that siUBE2F triggers downregulation of S6K1 total levels in PLC/PRF/5 cells, whereas in SK-HEP-1 cells siUBE2F has no effect on the levels of pS6K1 and p4E-BP1. The authors should clarify this discrepancy.

We apologize for some discrepancy. In some cases, the total-S6K1 levels were detected using stripped membrane after probing pS6K1. To address this critique, we have performed following experiments: 1) we optimized gel exposure time to show a significant inactivation of pS6K1 following UBE2F knockdown in SK-HEP-1 cells; 2) we have now re-run the gels and used the fresh blots to detect the levels of p4E-BP1 and S6K1, revealing marked inhibition of p4E-BP1 and minimal inhibition of S6K1; and 3) we quantified the band densities, and presented the results in the ratios of pS6K1/S6K1 and p4E-BP1/4E-BP1, which showed that UBE2F knockdown significantly reduced the phosphorylation of both S6K1 and 4E-BP1. These newly generated data are shown below and included in Figs. 2B and EV2F,G.

Figure EV2B. Similar issue is present in this figure. pS6K1 levels do not change in siUBE2F Hep3B cells. In addition, pS6K1 levels seem to be affected upon UBE2M depletion thus contradicting the hypothesis of the authors.

In particular, pS6K1 seems to be downregulated in siUBE2M Hep3B cells, whereas it seems to be upregulated in siUBE2M Sk-Hep-1 cells.

We agreed with the reviewer that there is indeed cell-line dependent minor discrepancy. In a shorter exposure film, we did observe a clearer pS6K1 inhibition upon UBE2F knockdown in Hep3B cells (shown now in Fig. EV2H). With regards to UBE2M knockdown, we observed moderate increase or decrease in the levels of pS6K1 and p4E-BP1 among three liver cancer cell lines and HeLa cells (we have now acknowledge this in the text Pg 8). Thus, UBE2M is unlikely a *bona fide* mTORC1 regulator. In contrast, upon UBE2F knockdown, the mTORC1 signal was consistently and robustly inactivated in all three liver cancer cell lines, which, therefore, became the focus of this study.

Figure 2E. Western blot for UBE2F and SAG should be included to assess whether they can interact (either directly or indirectly) with the mTORC active complex.

Agreed. We have now performed this assay, and the results showed that neither UBE2F nor SAG directly interacted with the active mTORC1 complex (below and Fig. 2E).

Figure EV2C. Co-immunoprecipitation experiments should be performed to confirm that mTORC1 does not associate with UBE2F-SAG.

Yes, we have now performed IB analysis to detect SAG and Cullin 5 (CUL5, a SAG binding protein) in these fractions. The results showed that SAG-CUL5 proteins were detected mainly in fractions 23-26, indeed separating from the active mTORC1 complex which was mainly detected in fractions 19-22 (see below and Fig. EV2I), indicating that SAG does not directly bind to the active mTORC1 complex. The earlier detection of SAG than UBE2F is due to the complex formation of SAG with CUL5, leading to larger molecular weight.

Figure EV2D. The results shown here seem to contradict the authors' claim: "In fact, we found that UBE2F knockdown did not significantly affect the levels of few upstream regulators of mTORC1, including HIF1 α , DEPTOR, TSC1, or TSC2, although REDD1 levels were reduced in a cell line-dependent manner (Fig. EV2D)" Indeed, besides REDD1, also HIF1, DEPTOR, and TSC1-2 appear to be downregulated in the siUBE2F cells. This result may suggest that UBE2F depletion may alter mTORC1 activity by affecting the levels of its upstream regulators. The authors should address this point.

Changes in these proteins were rather minor, as compared to inactivation of mTORC1. We reason that these minor reductions are likely attributable to the suppression of global protein translation resulting from mTORC1 inactivation. We have now acknowledged this in the text (Pg 9).

Figures 3B-D. The authors should confirm these results using UBE2F-SAG inactive

mutant.

Per reviewer's suggestion, we overexpressed WT-UBE2F, along with its enzymatic-dead mutant UBE2F-C116A, under the conditions of glucose or amino acid deprivation and then refeeding. The results showed that wild-type UBE2F, but not UBE2F-C116A mutant, is capable of activating S6K1 phosphorylation. This newly generated data is now shown below and included in Fig. EV3H and in the text (Pg 9-10). Note that SAG RING mutants were extremely unstable, so we were unable to perform this experiment.

Figures 4A-B. Western blot for UBE2F and mTOR should be included to check whether these proteins could form a complex.

As suggested, we included western blots for UBE2F and mTOR in this pull-down assay and found that under unstressed physiological condition, RHEB and SAG bind to each other, but neither SAG nor RHEB binds to UBE2F or mTOR in these liver cancer cell lines. These newly generated data are now shown below and included in Fig. 4A,B and in the text (Pg 10).

Figure 4C. GST-only should be included as negative control.

We repeated the assay with inclusion of GST as a negative control. The results showed that SAG binds to GST-RHEB, but not GST *in vitro* (see below and Fig. 4C).

Figures 4G-H. Neddylation assays should be analyzed probing the neddylated substrate with a NEDD8 antibody.

We have repeated these assays, and probed the blots with anti-NEDD8 Ab (see below and Fig. 4G,H).

Figure 6. Importantly, the results of the functional analyses shown in this figure should also include genetic or pharmacological inactivation of mTOR. This is extremely important to demonstrate that, indeed, manipulation of the UBE2F-SAG-RHEB pathway affects the biology of the cells via mTOR regulation.

Please be advised that in Fig. 6, we demonstrated that knockdown of UBE2F or SAG (to inactivate mTORC1) induced growth inhibition, which can be rescued by overexpressing WT-RHEB, but not RHEB mutant, indicating a causal role of RHEB neddylation. Since the mTORC1 signal is already inactivated by knockdown of UBE2F or SAG in this assay, no need to further inactivate mTORC1 via genetic or pharmacological approach. To address this critique, we have instead overexpressed UBE2F or SAG in three liver cancer cell lines, and found that in SK-HEP-1 and Hep3B cells, overexpression of either UBE2F or SAG activated mTORC1 activity and stimulated cell growth. Importantly, this effect can be rescued by rapamycin

treatment, suggesting that the growth-promoting effects of UBE2F and SAG are mediated through mTORC1 activation. These newly generated data are shown below and included in Figs. 6D-F and EV6B-D.

Figure 7I. Similarly, in this figure genetic or pharmacological inactivation of mTOR should be performed.

We have now overexpressed UBE2F or SAG in SK-HEP-1 cells with PTEN knockdown, and found that overexpression of either UBE2F or SAG upregulated SOX9 protein level, which was largely rescued by rapamycin treatment (shown below and included in Fig. EV7M, in the text Pg 17).

Figure 7 A-G. Quantification and statistical analyses should be included for all these

experiments. Without quantitative analyses it is difficult to assess the biological relevance of this pathway.

We have provided quantifications for original Fig. 7A&B (now shown as Fig. 7B,C.) For original Fig. 7C, we quantified the ratio of liver weight to body weight, now shown in Fig. EV7E. The quantification for original Fig. 7E were shown in Fig. EV7G,H.

Minor concerns:

Figures 4D and 4F. It is not clear what type of NEDD8 antibodies (NEDD8-cullins and NEDD8-exo-Rheb) have been used in this experiment. Authors should clarify.

In Fig. 4D, we used anti-FLAG antibody for immunoprecipitation, followed by anti-NEDD8 antibody to detect neddylation of FLAG-tagged RHEB. In Fig. 4F, we used anti-RHEB antibody for immunoprecipitation, followed by anti-NEDD8 antibody to detect neddylation of endogenous RHEB. We have now clarified these in Figure legends.

Figure 4H. RHEB-K169R mutant is used in this experiment without a previous explanation. Author should either reorganize the order of the figures or comment in the main text how they identified this mutant before figure 5. In the current version of the manuscript this mutant is described only afterwards, i.e., in figure 5.

Agreed. We have now modified the text to make it clear (Pg 11). Thank you.

Referee #2:

Summary:

In this manuscript, Zhang et al identified RHEB as a novel substrate for neddylation by the UBE2F-SAG E2-E3 axis, which modifies RHEB at the K169 site to enhance its lysosome localization and GTP-binding affinity, thereby activating mTORC1. They also demonstrated the role of UBE2F-SAG in regulating mTORC1 activity and its potential in treating liver diseases. Overall, this manuscript provides detailed mechanistic insights into how neddylation influences mTORC1 activity. This work is innovative and significant. Data set support the proposed hypothesis. Several major and minor points need to be addressed:

We deeply appreciate very positive comments and excellent summary of our work from this reviewer, and will address his/her critiques point-by-point below.

Major points:

1. In Figure 6, the author suggests that overexpression of RHEB could counteract mTORC1 inactivation, growth suppression, cell size reduction, and autophagy induction in UBE2F-knockdown cells. This implies that WT RHEB might still activate mTORC1 in the absence of proper neddylation. Further experiments are needed to clarify whether overexpressed WT RHEB can compensate for the loss of UBE2F and whether this activation occurs independently of neddylation.

It is well-known that RHEB is an activator of mTORC1. In Fig. 6, we used UBE2F knockdown cells instead of cells with UBE2F totally KO. It is possible that remaining

amount of UBE2F may still regulate RHEB activity and subcellular localization. It is worthy-noting that neddylation modification of any substrate is not up to 100%. Our experiments highly suggest that neddylated RHEB by the UBE2F-SAG axis is more active than wt-RHEB, but wt-RHEB remains an mTORC1 activator.

Nevertheless, to address this insightful question, we tried to generate UBE2F knockout cells using the Crisper system (sgUBE2F), followed by WT-RHEB overexpression. Unfortunately, among 29 individually selected clones, we did not obtain a single clone with complete UBE2F knockout in SK-HEP-1 liver cancer cells (below), further supporting our previous observation in lung cancer cells that UBE2F is growth essential for cancer cells. We are, therefore, unable to perform this suggested experiment.

2. What are the expression levels of RHEB under conditions of serum starvation and refeeding? Given that some studies suggest energy status may directly affect RHEB levels (<https://doi.org/10.1038/s41419-024-06808-1>), it is important to determine if there are any changes in RHEB expression.

Great point! We performed this suggested experiment with the citation of this important reference. Specifically, we examined RHEB levels in both liver cancer cell lines and MEF cells, and found that serum starvation followed by refeeding did not affect RHEB protein levels in all lines tested under our experimental condition. These newly generated data are shown below and included as Figs. 2D and EV3A.

3. To thoroughly evaluate the role of UBE2F in tumor cell proliferation, establishing

stable cell lines with UBE2F knockdown is recommended, particularly for clonogenic survival assays.

As suggested, we generated stable UBE2F knockdown cell lines and repeated the growth and clonogenic assays. Our results (shown below and included in Figs. 1D,E and EV1A,B) indicated that UBE2F knockdown effectively inhibits the growth and colony formation of HCC cells.

4. Similarly, the hypothesis testing in Figure 6 would benefit from the use of stable cell lines expressing the RHEB-K169R mutant. This approach would allow for a more consistent and controlled study of the mutant's effects.

We established stable cell lines using lentivirus-based technology and repeated growth and clonogenic assays. Consistent with our previous data, WT-RHEB, but not its neddylation deficient RHEB-K169R mutant, rescued the inhibition of growth and clonogenic survival induced by knockdown of either UBE2F or SAG (shown below and in Figs. 6A-C and EV6A).

5. The author notes that RHEB can undergo multiple post-translational modifications. Is there a potential interplay between neddylation and other modifications, such as ubiquitylation?

To address this thoughtful inquiry, we first used an *in vivo* polyubiquitylation assay, and found that RHEB-K169R mutant cannot be ubiquitylated, indicating that K169 is the site for both neddylation and ubiquitylation, and implying a potential competition between neddylation and ubiquitylation. We then determined whether UBE2F knockdown also affects RHEB ubiquitylation. The results showed that while UBE2F knockdown inhibits RHEB neddylation, it has no effect on RHEB ubiquitylation. This newly generated data is shown below and included in Appendix Fig. S2 and in the text (Pg 11).

6. The author mentions that targeting the UBE2F-SAG axis may have therapeutic advantages for certain liver diseases. Has the author tested whether the neddylation inhibitor MLN4924 could enhance the effectiveness of current anti-cancer therapies

in tumors with high UBE2F expression?

Great point and thank you! Our previous collaborative study had shown that general neddylation inhibitor MLN4924 inhibited growth of liver cancer cells both *in vitro* and *in vivo* by inducing autophagy (Cancer Res 2012, PMID: 22562464). Unfortunately, we have not yet performed such drug combination experiment in tumors with high UBE2F expression, which will certainly be an exciting project for future investigation. We have now discussed this important point in the Discussion section of this revision (Pg 22).

Minor Points:

1. Please ensure that the manuscript format is consistent throughout the document.

We have gone over the entire manuscript, and ensured the format is consistent throughout in this revised manuscript.

2. Ensure that the height of the western blot bands is consistent across all figures. For example, compare the band height in Figure 4A IP RHEB with IP SAG.

We have done this across all figures in this revised manuscript.

3. Please provide β -actin controls in all panels. Several panels are missing these controls, such as in Figure 4A compared to 4B.

We have now provided β -actin controls in all panels in this revised manuscript.

Referee #3:

In this paper, Zhang et al found that UBE2F, a Nedd8 (N8) E2 that works together with the E3 SAG/RBX2, Neddylates RHEB, leading to mTORC1 activation, increased cell size and proliferation (and inhibiting autophagy), promoting liver tumor growth. These authors thus identified RHEB as a novel substrate for UBE2F/SAG (different than the well-known Cul5 substrate). The Neddylation of RHEB leads to its lysosomal membrane localization and elevated GTP binding (ie active RHEB), important for activation of mTORC1 at the lysosomal membrane.

Overall, this is an interesting, extensive, systematic and carefully executed study that spans in vitro, in cellulo and in vivo studies (the latter using ube2f conditional knockout mice in the liver), as well as analysis of human liver tissues and tumors.

We thank this reviewer for his/her high level of enthusiasm, and excellent summary of our work.

Critique:

1. Much of the work presented relies on Western blots (WB) and IPs, which is logical, in order to study the biochemical processes involved. The major problem, however, is that none of these biochemical analyses (WB, IP) are quantified, with no statistics provided. Moreover, in most cases normalization of the data is required, but not provided. Eg pS6 needs to be normalized to total S6, and p4E-BP normalized to total 4E-BP). For example, in Fig 2C and Fig 3B,C: KO or KD or UBE2F or SAG resulted in reduction of p4E-BP, but so does total 4E-BP. So, is mTORC1 really inhibited by

lack of UBE2F?

Thank you for raising this critical issue. Yes, we have now performed the quantification of most relevant western blots, and calculated the ratio between the levels of phosphorylated form vs. total, as shown below and included in Fig. EV2A-D,F,G, and EV3B-E. We have also performed the statistical analysis for the ratios of pS6K1/S6K1 and p4E-BP1/4E-BP1 in Fig. 2A,B and Fig. 3B,C. Also see our response to Reviewer #1, concern on Fig. 2B.

2. Likewise, immunofluorescence (IF) studies of RHEB localization to lysosomes following Neddylation show only 1-2 cells, with no statistics provided (eg Pearson coefficient to prove colocalization with lysosomes)

So, overall, while these results appear compelling (based on one blot per experiment or 1-2 cells shown), lack of quantitation and statistics is problematic and makes it difficult to have confidence in some of the conclusions drawn.

We have performed the statistical analysis of the co-localization between RHEB and LAMP2 in HeLa and PLC/PRF/5 cells. The results showed that the lysosomal localization of the RHEB-K169R mutant was significantly decreased, compared to WT-RHEB. Furthermore, NEDD8 overexpression enhanced the co-localization of WT-RHEB with LAMP2, but did not affect the RHEB-K169R mutant. These newly generated data are shown below and included in Fig. EV5G,I.

3. *How does Neddyltion of RHEB target it to the lysosomal membrane? Is Neddyltion of Cul5 also targets it to the lysosomal membrane?*

It is well-known that neddyltion modification causes covalently attachment of NEDD8 to a substrate, which alters cellular localization of numerous substrates proteins. Given that RHEB K169R site is spatially proximal to its C-terminal farnesylation site (C181) for membrane targeting, we hypothesized that neddyltion may increase RHEB farnesylation, thereby enhancing its localization to lysosomes. To test this hypothesis, we generated RHEB-C181S, a well-known RHEB un-farnesylated mutant (Oncogene 2010, PMID: 19838215). Using the immunofluorescence staining, we found that both K169R and C181S mutants displayed a similar dispersed subcellular distribution (shown below and included in Appendix Fig. S3), highly suggesting that neddyltion may influence RHEB farnesylation. We further used immunoprecipitation assay to examine the farnesylation levels of WT-RHEB and RHEB-K169R mutant. The results showed that wild-type RHEB is farnesylated, which is inhibited by FTI-277, a farnesylation inhibitor, whereas the farnesylation level of RHEB-K169R mutant is significantly reduced (shown below and included in Fig. 5D). Thus, it appears that RHEB neddyltion promotes its lysosome localization by enhancing its farnesylation. Finally, while determining whether Cul5 neddyltion also targets cullin-5 to lysosomal membrane is of interest, it is certainly out of the scope of this investigation, and we hope that the reviewer will agree with us. Thank you.

4. *And how does Neddyltion of RHEB lead to increase of its GTP binding (active form)?*

Another great question, and we have tried to address this. As shown in Fig. 5F, we

performed molecular dynamics (MD) simulation analysis, and found that the GTP-binding loop of unmodified RHEB has a trend to swing out, while neddylated RHEB preserves a stable conformation in the GTP-binding loop which forms a hydrogen bond between RHEB and GTP, leading to an increased GTP binding affinity, highly suggesting that neddylation of RHEB may stabilize its binding with GTP. We have now included this explanation in the text (Pg 14).

5. Fig EV4D (Neddylation of RHEB in 293 cells): the faint band shown to represent Neddylation is not convincing, especially as it is not reduced in the presence of the Neddylation inhibitor MLN4924.

We have repeat this assay, using light chain-free secondary antibody to avoid interference from heavily abundant IgG (shown in Fig. EV4D).

6. How does UBE2F promote cell proliferation without affecting the cell cycle?

Another great question! Given the fact that we did not observe significant change in cell cycle progression by regular FACS analysis, but did observe significant change in cell size upon UBE2F knockdown in these liver cancer cells, we should have concluded that UBE2F mainly regulates cell growth (not cell proliferation) and autophagy via manipulating mTORC1 activity. To completely rule in or rule out the effect of UBE2F knockdown on cell cycle progression of liver cancer cells, we have now used sensitive BrdU-incorporation assay, and found that UBE2F knockdown significantly slowed the S phase entry. We further used double thymidine block assay to synchronize cells at the G1/S boundary of the cell cycle. The results from the release of double thymidine block revealed that UBE2F knockdown caused cell arrest at both G1 and G2/M phases of cell cycle with increased levels of p21 (for G1 phase) and cyclin B1 (for G2/M phase). These newly generated data are shown below and included in Figs. 1F, EV1C-F and Appendix Fig. S1D-H. Based upon these results, we concluded that UBE2F knockdown inhibits both cell growth and cell proliferation in addition to autophagy induction, as a result of mTORC1 inactivation. We have modified our results in the text to reflect these changes (Pg 6-7).

7. Analysis of liver tumor tissues: were they graded blindly?

Given they were not graded blindly in our original submission; we have collaborated with a senior investigation whose lab members have pathological expertise in liver cancer. The staining of liver tumor tissue microarray samples was re-scored and the data were shown below and included in Figs. 1B,C, 7L,M and Appendix Fig. S1C, This investigator is now included as a co-author in this revision.

Dear Dr Sun,

Thank you for submitting your revised manuscript (EMBOJ-2024-118062R) to The EMBO Journal, as well for your patience with our response. Your amended study was sent back to the three referees for their scientific re-evaluation, and we have received detailed comments from all of them, which I enclose below. As you will see, the experts state that the work has been substantially improved by the revisions and they are now broadly in favour of publication.

Thus, we are pleased to inform you that your manuscript has been accepted in principle for publication in The EMBO Journal.

We now need you to take care of a number of issues related to formatting and data presentation as detailed below, which should be addressed at re-submission.

Please contact me at any time if you have additional questions related to below points.

As you might have seen on our web page, every paper at the EMBO Journal now includes a 'Synopsis', displayed on the html and freely accessible to all readers. The synopsis includes a 'model' figure as well as 2-5 one-short-sentence bullet points that summarize the article. I would appreciate if you could provide this figure and the bullet points.

Thank you for giving us the chance to consider your manuscript for The EMBO Journal. I look forward to your final revision.

Again, please contact me at any time if you need any help or have further questions.

Best regards,

Daniel Klimmeck

>> Please limit the number of keywords for your study to maximally five.

>> Author Contributions: Please remove the author contributions information from the manuscript text. Note that CRediT has replaced the traditional author contributions section as of now because it offers a systematic machine-readable author contributions format that allows for more effective research assessment. and use the free text boxes beneath each contributing author's name to add specific details on the author's contribution.

More information is available in our guide to authors.
<https://www.embopress.org/page/journal/14602075/authorguide>

>> Funding: enter the following funding information in the list of funders in our online system: - 32471300 (MOST).

>> Rename the current 'Conflict of Interests' section to 'Disclosure and Competing Interests Statement'.

>> Add a Reagents and Tools table to the Methods section, listing key reagents, experimental models, software and relevant equipment.

>> Data availability section: Add a statement 'No data amenable to large-scale data deposition were generated in this study.' .

>> As to our journal policies we kindly ask you to check & clarify overlap of data within Fig.1i .

>> Consider additional changes and comments from our production team as indicated below:

DATA CHECK: PASS

- Figure legends:

1. Please note that the exact p values are not provided in the legends of figures 1b, d-f, j-k; 6a c-d, f, l; 7b-c; EV 1a-b, d-e; EV 2a-d, f-g; EV 3b-e; EV 5g, i; EV 6a-b; EV 7b, d-e, g-h, k.
2. Please indicate the statistical test used for data analysis in the legends of figures 1c; 7l-m.
3. Please note that information related to n is missing in the legend of figure 1b.
4. Please note that the error bars are not defined in the legend of figure 1b.
5. Please note that the red dotted box is not defined in the legend of figure 5a. This needs to be rectified.
6. Please note that the orange arrows are not defined in the legend of figure EV 1h. This needs to be rectified.

Referee #1:

The authors have been highly responsive to previous critiques. The revision has addressed all my concerns and is much improved.

Referee #2:

The authors have provided additional data that effectively address my concerns. I have no further reservations and recommend this manuscript for publication in the EMBO Journal.

Referee #3:

The authors have now addressed my previous concerns by providing more data and better quantitative analyses of their results. I do not have further concerns.

The authors addressed the remaining editorial issues.

Dear Dr Sun,

Thank you for submitting the revised version of your manuscript. I have now evaluated your amended manuscript and concluded that the remaining minor concerns have been sufficiently addressed.

I am thus pleased to inform you that your manuscript has been accepted for publication in the EMBO Journal.

Related I would like to hereby ask your consent on keeping the referee figures included in this file.

On a different note, I would like to alert you that EMBO Press offers a format for a video-synopsis of work published with us, which essentially is a short, author-generated film explaining the core findings in hand drawings, and, as we believe, can be very useful to increase visibility of the work. Please see the following link for representative examples and their integration into the article web page:

<https://www.embopress.org/doi/full/10.15252/emj.2019103932>

Best regards,

Daniel Klimmeck

Daniel Klimmeck, PhD
Senior Editor
The EMBO Journal
EMBO
Postfach 1022-40
Meyerhofstrasse 1
D-69117 Heidelberg
contact@embojournal.org
Submit at: <http://emboj.msubmit.net>
